Manuscript prepared for Earth Surf. Dynam.
with version 2014/09/16 7.15 Copernicus papers of the LaTeX class copernicus.cls.
Date: 20 July 2016

# The CAIRN method: Automated, reproducible calculation of catchment-averaged denudation rates from cosmogenic nuclide concentrations

Simon Marius Mudd[1], Marie-Alice Harel[1], Martin D. Hurst[2], Stuart W. D. Grieve[1], and Shasta M. Marrero[1]

[1]School of GeoSciences, University of Edinburgh, Drummond Street, Edinburgh EH8 9XP, UK
[2]British Geological Survey, Keyworth, Nottingham NG12 5GG, UK

*Correspondence to:* Simon M. Mudd (simon.m.mudd@ed.ac.uk)

**Abstract.** We report a new program for calculating catchment-averaged denudation rates from cosmogenic nuclide concentrations. The method (Catchment-Averaged denudatIon Rates from cosmogenic Nuclides: CAIRN) bundles previously reported production scaling and topographic shielding algorithms. In addition, it calculates production and shielding on a pixel-by-pixel basis. We explore the sampling frequency across both azimuth ($\Delta\theta$) and altitude ($\Delta\phi$) angles for topographic shielding and show that in high relief terrain a relatively high sampling frequency is required, with a good balance achieved between accuracy and computational expense at $\Delta\theta = 8°$ and $\Delta\phi = 5°$. The method includes both internal and external uncertainty analysis, and is packaged in freely available software in order to facilitate easily reproducible denudation rate estimates. CAIRN calculates denudation rates but also automates catchment averaging of shielding and production, and thus can be used to provide reproducible input parameters for the CRONUS family of online calculators.

## 1 Introduction

*In-situ* cosmogenic nuclides, such as $^{10}$Be and $^{26}$Al, are widely used to determine both exposure ages and denudation rates (e.g., Dunai, 2010; Granger et al., 2013; von Blanckenburg and Willenbring, 2014; Granger and Schaller, 2014). A denudation rate is the sum of the chemical weathering rate and physical erosion rate. Since the publication of the seminal papers by Brown et al. (1995), Granger et al. (1996) and Bierman and Steig (1996), dozens of studies have used concentrations of cosmogenic nuclides in stream sediments to quantify denudation rates that are spatially averaged over eroding drainage basins. There are now more than 1000 published catchment-averaged denuda-

tion rates (e.g., Portenga and Bierman, 2011; Willenbring et al., 2013a; Harel et al., 2016), with many new studies published each year.

Several authors have provided standardized methods for calculating denudation rates from cosmogenic nuclide concentrations, notably the COSMOCALC package (Vermeesch, 2007) and the CRONUS-Earth online calculator (Balco et al., 2008). Here we make comparisons with the CRONUS
calculator version 2.2, so we refer to it as CRONUS-2.2 for clarity. These calculators have been widely adopted by the cosmogenic, quaternary science and geomorphic communities, in large part because they are easily accessible and their methods are transparent (i.e., the source files are available online). These previously published calculators are ideal for calculating denudation rates or ages from a particular site (e.g., an exposed surface or a glacial moraine). Existing calculators rely
on the principle that there is an inverse relationship between denudation rate and the concentration of a nuclide, because slower denudation results in more exposure to cosmic rays. In addition, these calculators make use of the fact that the concentration of a nuclide can be inverted for denudation rate if one estimates the production of the nuclide.

In the context of catchment-averaged denudation rates, nuclide production rates will vary in space,
and an open-source method of calculating production and inverting nuclide concentration for denudation rate has yet to emerge. Due to the lack of an open-source tool, a wide variety of approaches to calculating catchment-averaged denudation rates are used in the literature, which makes intercomparison studies challenging (cf., Portenga and Bierman, 2011; Willenbring et al., 2013a; Harel et al., 2016).

Several factors determine the concentration of a cosmogenic nuclide in a sample. For instance, elevation and latitude control the production rate of different cosmogenic nuclides (e.g., Lal, 1991; Dunai, 2000; Stone, 2000; Desilets and Zreda, 2003; Lifton et al., 2005). Production rates vary spatially, thus users of online calculators must calculate the effective production rate within a catchment using a weighted mean of nuclide production in individual pixels. The manner in which these are
provided to existing calculators vary. For example, one must feed a single weighted mean production, after shielding corrections, to COSMOCALC. In contrast, one must calculate weighted mean shielding corrections and pass them to CRONUS-2.2, and in addition must calculate a pressure or elevation that reproduces the mean production rate before shielding.

Many authors use an averaging scheme for production wherein production is calculated in each
pixel which is then passed to a calculator (e.g., Kirchner et al., 2001; Hurst et al., 2012; Munack et al., 2014; Scherler et al., 2014). In addition, nuclide concentrations can be affected by partial shielding caused by snow cover, surrounding topography, and overlying layers of sediment (e.g., Balco et al., 2008). These again are spatially distributed and so authors reporting catchment-averaged denudation rates frequently report averaged shielding values. Although software packages do exist
for calculating spatially averaged topographic shielding (e.g., Codilean, 2006) and snow shielding (e.g., Schildgen et al., 2005), results from these models are not integrated with spatially varying

production rates. Finally, in landslide dominated terrain, removal of thick layers of sediment can dilute cosmogenic nuclide concentrations in river sediment (Niemi et al., 2005; Yanites et al., 2009; West et al., 2014). This factor is often not included in denudation calculations. For these reasons, Balco et al. (2008) specifically urged development of tools dedicated to the calculation of catchment-averaged denudation rates from cosmogenic nuclide concentrations.

Here we present software that estimates production and shielding of the cosmogenic nuclides [10]Be and [26]Al on a pixel-by-pixel basis, and propagates uncertainty in AMS measurement and cosmogenic nuclide production. Based on these calculations the software can then calculate the expected cosmogenic nuclide concentration from a basin given a spatially homogenous denudation rate. Finally, the software uses Newton iteration to calculate the denudation rate that best reproduces the measured cosmogenic nuclide concentration. We have made this software available through an open-source platform at https://github.com/LSDtopotools/LSDTopoTools_CRNBasinwide to allow community modification and scrutiny, with the goal of enabling users to report denudation rates that can be easily reproduced by other scientists. The software distribution includes instructions for building the software on a virtual machine that can function on common operating systems.

## 2 Quantifying denudation rates at a single location

We derive a governing equation that tracks the concentration of a cosmogenic nuclide as it is exposed, exhumed or buried. This approach is adopted because it is the most general: specific scenarios of both steady and transient denudation and burial may therefore be derived. Our approach is broadly similar to that of Parker and Perg (2005), but results are equivalent to those of more widely used derivations (e.g., Lal, 1991; Granger and Smith, 2000).

We begin by conserving the concentration of cosmogenic nuclide $i$ through time $t$:

$$\frac{dC_i}{dt} = P_i - \lambda_i C_i \tag{1}$$

where $C_i$ is the concentration of cosmogenic nuclide $i$ ($C_i$ is typically reported in atoms g[-1]; $i$ could be [10]Be or [26]Al, for example), $P_i$ is the local production rate of cosmogenic nuclide $i$ (in atoms g[-1] yr[-1]) and $\lambda_i$ (yr[-1]) is the decay constant of cosmogenic nuclide $i$. Production can be a function of latitude, altitude (or atmospheric pressure), magnetic field strength and shielding by rock, soil, water or snow (e.g., Balco et al., 2008).

Cosmogenic nuclides can be produced by both neutrons and muons (e.g., Gosse and Phillips, 2001). Production by neutrons is widely modelled using a simple function in which production decays exponentially with depth (e.g., Lal, 1991). Muons, on the other hand, are modelled using a variety of schemes. The CRONUS-2.2 calculator (Balco et al., 2008) implements the scheme of Heisinger et al. (2002a, b), which requires computationally expensive integration of muon stopping over a depth profile. Field-based estimates of muon production demonstrate that Heisinger et al.

(2002a) significantly overestimate production by muons (Braucher et al., 2011, 2013; Phillips et al., 2016a). Other authors have used empirical fits of cosmogenic profiles from the field, typically using a sum of exponential functions, to describe muon production (e.g., Granger and Smith, 2000; Vermeesch, 2007; Braucher et al., 2009; Schaller et al., 2009).

The advantage of the Heisinger et al. (2002a) scheme is that it tries to capture the physics of muon passage through the near surface, and specifically the scheme models how the mean energy of muons increases as one moves to greater depths in the subsurface. This affects muon production at depth in a way that is not captured by exponential approximations. Recent work by Marrero et al. (2016) has updated the scheme of Heisinger et al. (2002a, b) to reflecting the muon production rates inferred
from field studies. This method still has the disadvantage that it is computationally expensive, to the extent that this computational cost is prohibitive if one is to calculate muon production in numerous pixels across a catchment.

Our approach is to approximate muon production using a sum of exponential functions (e.g., Granger and Smith, 2000; Vermeesch, 2007; Braucher et al., 2009; Schaller et al., 2009). This
approach has the advantage of being computationally efficient, but it does not reflect the physics of muon production and therefore does poorly at capturing muon production at depths beyond a few meters. This is unlikely to lead to large errors, however, because muon production makes up a very small percentage of the overall nuclide production at the depths where the physics-based models (Heisinger et al., 2002a, b; Marrero et al., 2016) diverge from the exponential models used in
CAIRN. We specifically quantify this difference in Section 6.3, finding that the exponential approximation leads to differences between the physics-based approximation that are relatively small: for a wide range of denudation rates these differences are less than 2%.

The exponential approximation for nuclide production used in CAIRN is:

$$P_i(d) = P_{i,SLHL} \sum_{j=0}^{3} S_{i,j} F_{i,j} e^{\frac{-d}{\Lambda_j}} \tag{2}$$

where $P_{i,SLHL}$ is the surface production rate (atoms g$^{-1}$ yr$^{-1}$) at sea level and high latitude, $F_{i,j}$ is a dimensionless scaling that relates the relative production of neutron spallation and muon production, $S_{i,j}$ is a dimensionless scaling factor that lumps the effects of production scaling and shielding of cosmic rays, $d$ is a mass per unit area which represents the mass overlying a point under the surface (typically reported in g cm$^{-2}$), and $\Lambda_j$ is the attenuation length for reaction type $j$ (g cm$^{-2}$). The
reaction types are $j = 0$ for neutrons and $j = 1 - 3$ for muons; muons can be either slow or fast. In general, production from muons relative to neutrons is greater in landscapes with a high denudation rate or at low elevation (Balco et al., 2008).

The depth $d$, called shielding depth, is related to depth below the surface as:

$$d = \int_{\zeta-\eta}^{\zeta} \rho(z)\,\mathrm{d}z \tag{3}$$

where $\zeta$ (cm) is the elevation of the surface, $\eta$ (cm) is the depth in the subsurface of the sample, $z$ (cm) is the elevation in a fixed reference frame and $\rho$ (g cm$^{-3}$) is the material density, which may be a function of depth. For a constant density, $d = \rho\eta$.

## 2.1  Solving the governing equation

The governing equation (Eq. 1) has the general form:

$$\frac{dC}{dt} + p(t)C = g(t) \tag{4}$$

In our case, $p(t)$ simply equals $\lambda_i$, which is a constant in this case, and $g(t)$ is equal to $P_i$, which is a function of $t$.

Equations of this form have the solution:

$$C = \frac{1}{h(t)} \int h(t)g(t)\,\mathrm{d}t + const \tag{5}$$

where $const$ is an integration constant and

$$h(t) = \exp\left(\int p(t)\,\mathrm{d}t\right) \tag{6}$$

which in the case of the governing equation reduces to:

$$h(t) = e^{\lambda_i t} \tag{7}$$

The term $g(t)$ is equal to:

$$g(t) = P_{i,SLHL} \sum_{j=0}^{3} S_{i,j} F_{i,j} e^{\frac{-d}{\Lambda_j}} \tag{8}$$

The shielding depth, $d$, is a function of time:

$$d(t) = d_0 + \int_{t_0}^{t} \epsilon(\tau)\mathrm{d}\tau \tag{9}$$

where $\tau$ is a dummy variable for time that is replaced by the limits after integration. Here $t_0$ is the initial time and $d_0$ is the initial shielding depth. In the case where denudation, denoted $\epsilon$ (g cm$^{-2}$ yr$^{-1}$), is steady in time this becomes:

$$d(t) = d_0 + \epsilon(t_0 - t) \tag{10}$$

Here denudation is the rate of removal of mass from above the sample per unit area. If we let the concentration of the cosmogenic nuclide equal $C_0$ at the initial time, $t_0$, and combine Eqs. (5), (7), (8), and (10), we can solve for the integration constant ($const$) and arrive at a solution for cosmogenic nuclide $i$ at time $t$:

$$C_i(t) = C_0 e^{-(t-t_0)\lambda_i} + P_{i,SLHL}\left[ \sum_{j=0}^{3} \frac{S_{i,j}F_{i,j}\Lambda_{i,j}}{\epsilon + \Lambda_{i,j}\lambda} e^{\frac{-d_0}{\Lambda_{i,j}}} \left( e^{\frac{\epsilon(t-t_0)}{\Lambda_{i,j}}} - e^{-(t-t_0)\lambda} \right) \right] \tag{11}$$

Equation (11) is the full governing equation from which scenario-specific solutions may be derived.

## 2.2 Steady state solution

By convention, we consider the depth profile of cosmogenic nuclide concentration to be steady in time. This allows analytical solution of the cosmogenic nuclide concentration at any point in the basin. At steady state, the particles near the surface have been removed (either through erosion or chemical weathering) at the same rate for a very long time, so we set $t_0 = 0$ and $t = \infty$. This results in a simplified form:

$$C_i(d) = P_{i,SLHL} \sum_{j=0}^{3} \frac{S_{i,j}F_{i,j}\Lambda_{i,j}e^{-d/\Lambda_{i,j}}}{\epsilon + \lambda_i\Lambda_{i,j}} \tag{12}$$

where $\epsilon$ is the denudation rate (g cm$^{-2}$ yr$^{-1}$). If we set $d = 0$ (that is, we solve for material being eroded from the surface, with no distributed mass loss via chemical weathering), Eq. (12) reduces to Eq. (6) from Granger and Smith (2000) for denudation only (i.e., no burial or exposure), and reduces to Eq. (8) of Lal (1991) if production is due exclusively to neutrons. If Eq. (12) is simplified to neutron only production, assumes the sample is taken from the surface ($d = 0$), and is solved for erosion rate, one arrives at

$$\epsilon = \Lambda_i \left( \frac{P_{i,SLHL}S_i}{C_i} - \lambda_i \right) \tag{13}$$

which is equivalent to the widely used Eq. (11) from Lal (1991). However Eq. (13) requires adjustment for catchment averaged estimates of denudation rates because each point in the landscape from which sediment is derived will have its own local production and shielding factors. This is why a spatially distributed approach is required.

## 2.3 Snow and self shielding

Equation (12) is restrictive in that it only considers material removed from a specific depth, i.e. removed for a single value of $d$. In reality samples may come from a zone of finite thickness. This finite thickness can contribute some shielding to the sample, i.e. the bottom of a sample is shielded by the mass of the sample that overlies. This shielding is called self shielding and is generally implemented by assuming that self shielding can simply be approximated by a reduction in neutron production (e.g., Vermeesch, 2007; Balco et al., 2008). Snow can also reduce production of cosmogenic nuclides (e.g., Gosse and Phillips, 2001). Typically these two forms of shielding (snow and self) are incorporated in denudation rate calculators as a scaling coefficient calculated before solving the governing equations (e.g., Vermeesch, 2007; Balco et al., 2008), i.e. snow and self shielding are incorporated into the $S_{i,j}$ term.

Our strategy is slightly different: we calculate snow and self shielding by integrating the cosmogenic nuclide concentration over a finite depth in eroded material. For example, if there is no snow, the concentration of cosmogenic nuclides at a given location is obtained by depth-averaging the steady concentrations from zero depth (the surface) to the thickness of eroded material. If snow is present, the concentration is determined by depth-averaging from the mean snow depth ($d_s$) to the thickness of the removed material ($d_t$). Both $d_s$ and $d_t$ are shielding thicknesses, therefore they are in units of g cm$^{-2}$ and thus differences in material density are taken into account. The depth-averaged concentration is then:

$$C_i(d) = \frac{P_{i,SLHL}}{d_t} \sum_{j=0}^{3} \frac{S_{i,j} F_{i,j} \Lambda_{i,j}^2 \left(e^{-d_s/\Lambda_{i,j}} - e^{-(d_s+d_t)/\Lambda_{i,j}}\right)}{\epsilon + \lambda_i \Lambda_{i,j}} \tag{14}$$

In most applications, the thickness of the removed material will be 0, i.e. the particles from which nuclide concentrations are measured in detrital sediment are derived from a thin layer removed from the surface of the catchment. However, the solution described by Eq. (14) allows some flexibility so that future users can explore different erosion scenarios, for example removal of sediment through mass wasting. We discuss this in Sect. 5.4, but for the current contribution we focus on steady-state scenarios.

## 2.4 Topographic shielding

In addition to snow and self shielding, locations in hilly or mountainous areas can also receive a reduced flux of cosmic rays because these have been shielded by surrounding topography (Dunne et al., 1999). We adopt the method of Codilean (2006), in which both the effect of dipping sample surfaces and shielding by topography blocking incoming cosmic rays are computed. The Codilean (2006) method is spatially distributed: each pixel in a digital elevation model (DEM) has its own topographic shielding correction that varies from 0 (completely shielded) to 1 (no topographic shield-

ing). These correction values are calculated by modelling shadows cast upon each pixel in the DEM

from every point in the sky. This is achieved by modelling shadows incrementally for a range of zenith ($\phi$) values from $0°$ to $90°$ and azimuth ($\theta$) values from $0°$ to $360°$.

As $\Delta\theta$ and $\Delta\phi$ values decrease, the accuracy with which the shielding is calculated is expected to increase, as we are modelling shielding at finer resolutions. However, this benefit is attenuated by increasing computational cost when these values tend towards $(1°, 1°)$. Codilean (2006) compared

the accuracy of different $\Delta\theta$ and $\Delta\phi$ by comparing them to a minimum step size of $(5°, 5°)$. Here we exploit the efficiency of our software and the considerable increase in computing power since 2006 to explore smaller step sizes. We make the assumption that a step size of $(1°, 1°)$, corresponding to $32\,400$ iterations of the shielding algorithm, is an accurate representation of the true shielding factor to the extent that any further refinement in the measurements would not yield a significant change in

the results of the cosmogenic nuclide calculations.

In order to determine the optimal balance between measurement accuracy and computational efficiency, the full range of $(\Delta\theta, \Delta\phi)$ pairs were used to derive shielding values for each cell of a worst-case scenario: a high-relief section of the Himalaya (650 km$^2$ with a 7000 m range in elevation). Table 1 presents the maximum absolute residual value (the error of the pixel with the greatest

error) for topographic shielding of the corresponding step sizes when compared to the shielding derived for $(1°, 1°)$. Using values below Codilean (2006)'s suggested threshold of $(5°, 5°)$ gives increasingly small returns for a larger computational burden. We suggest that a $(\Delta\theta, \Delta\phi)$ pair of $(8°, 5°)$, requiring 810 iterations, is an optimal value for any high relief landscape, yielding a maximum absolute error in our test site of 0.018. On lower relief landscapes the $(\Delta\theta, \Delta\phi)$ values could be

increased to achieve the same level of accuracy. We note that these data are determined using a 90 m resolution DEM, and errors will be higher for finer resolution DEMs (Norton and Vanacker, 2009).

Our topographic shielding calculations rely on two approximations that can lead to some uncertainty. First, the method of Codilean (2006) assumes the horizon attenuates all cosmic rays, and secondly the production of cosmogenic nuclides obeys a power law relationship between the cosine

of the zenith angle. Argento et al. (2015) have shown these assumptions to be inaccurate. In addition, the Codilean (2006) method does not include changes to the flux penetration distance on the gradient of the topographic surface (e.g., Dunne et al., 1999; Balco, 2014). Thus our method, while precise, reflects a simplified model of the true physics of topographic shielding.

## 2.5 Production scaling

Production of cosmogenic nuclides varies as a function of both elevation (defined via atmospheric pressure) and latitude and these variations are accounted for by using one of several possible scaling schemes. The classic scaling model of Lal (1991), later modified by Stone (2000), is the simplest and is referred to herein as Lal/Stone. Later scaling models (Dunai, 2000, 2001; Desilets and Zreda, 2003; Lifton et al., 2005, 2014) have incorporated other parameters such as time-dependent geomag-

netic field variations, solar modulation, and nuclide-specific information, resulting in a total of seven
possible scaling models in the most recent CRONUS calculator (Marrero et al., 2016).

These scaling schemes vary in complexity and therefore computational expense. Time-dependent
scaling schemes are far more computationally expensive than the time-independent scheme of Lal/Stone,
which does not consider variations in geomagnetic field strength. Recent calibration results (Borchers
et al., 2016; Phillips et al., 2016a), including a low-latitude, high-altitude site in Peru (Kelly et al.,
2015; Phillips et al., 2016b) suggest that the time-independent Lal/Stone scheme performs similarly
to the physics-based schemes presented in Lifton et al. (2014) and fits the data better than several
other scaling schemes (Dunai, 2000; Desilets and Zreda, 2003; Lifton et al., 2005). For these reasons,
we scale production rates using the Lal/Stone scheme. This may lead to some uncertainty because
production rates are scaled by the intensity of the Earth's geomagnetic field (e.g., Dunai, 2010), and
this intensity has been relatively high over the last 20 kyrs (Valet et al., 2005; Lifton et al., 2014),
meaning that this approximation could lead to some uncertainty in samples with slow denudation
rates. For example, a rock removal rate of 0.03 mm/yr would remove 60 cm in 20 kyrs, and most
of production of nuclides occurs in the top 60 cm of rock (Lal, 1991). However, in cases with faster
denudation rates, the uncertainty introduced by assuming time-invariant production rates is likely to
be much smaller than other sources of uncertainty.

The Lal/Stone scaling scheme requires air pressure, whereas most published studies include only
elevation information. We follow the approach of Balco et al. (2008) and convert latitude and eleva-
tion data to pressure using the NCEP2 climate reanalysis data (Compo et al., 2011). In certain areas,
the ERA-40 reanalysis (Uppala et al., 2005) has been shown to provide more accurate results and
due to CAIRN's open source design new models can be readily incorporated into the software. Here
we retain the NCEP2 reanalysis to better compare our results with CRONUS-2.2. We note that if
users deploy CAIRN as a spatial averaging front end to online calculators, they should be vigilant to
use the same air pressure conversion method in both CAIRN and the online calculator.

**2.6 Combining scaling and shielding**

To calculate the concentration of a cosmogenic nuclide, the scaling factors for each production
pathway ($S_{i,j}$) must be computed. Both topographic shielding and production rate scaling are sub-
sumed within the scaling terms ($S_{i,j}$), whereas snow and self shielding are computed separately (see
Sect. 2.3). These scaling terms are not computed for each production pathway, but rather are lumped
into a single value. We therefore need to compute the values of the individual scaling factors, $S_{i,j}$.
To do this, we follow the method of Vermeesch (2007) and calculate scaling factors using an effec-
tive attenuation depth. This is necessary because, when considering multiple production pathways,
the scaling terms for individual production mechanisms may vary depending on elevation, shielding,
sample thickness, or denudation rates. For example, muogenic pathways will contribute relatively

more to production when there is more shielding since muogenic reactions penetrate deeper than
spallation.

To determine the scaling terms for the individual production mechanisms ($S_{i,j}$), we first compute the total scaling at a location ($S_{tot}$), which we define as the product of the production rate scaling ($S_p$) and the topographic shielding ($S_t$), that is $S_{tot} = S_t S_p$. Production scaling ($S_p$) is estimated

using the Lal/Stone scaling scheme and $S_t$ is calculated using our topographic shielding algorithms. We then derive the scaling factors for the individual production mechanisms, $S_i, j$, by employing a virtual attenuation length, $\Lambda_v$, in units of g cm$^{-2}$, following the method of Vermeesch (2007):

$$S_{i,j} = e^{\frac{-\Lambda_v}{\Lambda_i}} \tag{15}$$

We must therefore calculate $\Lambda_v$ based on $S_{tot}$. The individual production mechanisms must be set

such that:

$$S_{tot} = \sum_{j=0}^{3} S_{i,j} F_{i,j} \tag{16}$$

In Eq. (16), $S_{tot}$ and $F_{i,j}$ are known, whereas $S_{i,j}$ are functions of $\Lambda_v$. We thus iterate upon $\Lambda_v$, calculating $S_{i,j}$ using Eq. (15) using Newton's method until Eq. (16) converges on a solution for $\Lambda_v$. Once the virtual attenuation length is solved, the $S_{i,j}$ terms are then used in Eq. (14).

## 3  Denudation rates across a catchment

So far we have described the calculations that predict the concentration of a cosmogenic nuclide at one specific location in a basin. All existing cosmogenic nuclide calculators contain some form of these calculations. A wide variety of approaches to scale calculations of cosmogenic nuclide concentrations within a single location to the concentration across entire catchments have been used

in the literature. Some authors have averaged production rates on a pixel-by-pixel basis but have not considered topographic shielding (e.g., Belmont et al., 2007; DiBiase et al., 2010; Portenga and Bierman, 2011). Others have calculated an average scaling by integrating the product of topographic shielding and production on a pixel-by-pixel basis (e.g., Ouimet et al., 2009; Hurst et al., 2012; Lupker et al., 2012; Scherler et al., 2014). Another strategy is to calculate both averaged topographic

shielding and production scaling values for a basin (e.g., Abbühl et al., 2010). All of these approaches involve some degree of spatial averaging of production, shielding, or a combination of the two before catchment-averaged denudation rates can be estimated.

The approach we take in CAIRN differs in that shielding and production rates are not averaged: these are calculated locally at each pixel. For a given denudation rate, $\epsilon$, the concentration of cos-

mogenic nuclides from each pixel is calculated, then the catchment-averaged concentration is the

average of the concentrations from all pixels. This concentration requires no weighting because the denudation rate is considered to be spatially homogenous. The denudation rate for the basin is then iterated upon with Newton's method until the predicted concentration of cosmogenic nuclides emerging from the catchment matches the measured concentration (see Algorithm 1).

We should note here that the version of CAIRN reported in this contribution calculates the denudation rate across an entire catchment required to produce the observed concentration of the target cosmogenic nuclide. That is, CAIRN assumes denudation rates are the same everywhere in the catchment. Users can explore the effect of instantaneously removing mass by setting $d_t$ Eq. 14 and $d_t$ can be spatially heterogeneous, but even when users choose this option CAIRN will still calculate the

spatially homogenous background denudation rate in light of dilution by mass wasting or stripping of material from the landscape. Future adaptations of the code could account for nested basins, as this sampling strategy common in many studies of basin averaged erosion rates, or changes in the concentration of target minerals as employed by, for example Safran et al. (2006) or Carretier et al. (2015). Our software is open source so other groups can make adjustments to CAIRN to suit their

needs. These potential future developments, however, are beyond the scope of this contribution.

## 4    Uncertainty propagation

We calculate uncertainty from both internal (nuclide concentration uncertainties from accelerator mass spectrometry (AMS) measurements) and external (shielding and production rate) sources using Gaussian propagation of uncertainty following Balco et al. (2008). We do note that some authors

have used a Monte Carlo approach in determining cosmogenic nuclide-derived denudation rates because parameter uncertainties can have non-gaussian distributions (e.g., West et al., 2015). CAIRN, at present, does not implement a Monte Carlo uncertainty approach but rather follows conventional Gaussian propagation of uncertainty.

### 4.1    Gaussian propagation of uncertainty

Uncertainties are calculated in terms of the denudation rate, $\epsilon$, in units of g cm$^{-2}$ yr$^{-1}$, so that no assumption about material density is necessary. The standard deviation of the denudation rate, $s_\epsilon$, is calculated with

$$s_\epsilon = \sqrt{\left(\frac{\partial \epsilon}{\partial x}\right)^2 s_x^2 + \left(\frac{\partial \epsilon}{\partial y}\right)^2 s_y^2 + \dots} \qquad (17)$$

where $s_x$ is the standard deviation of $x$, $s_y$ is the standard deviation of $y$, and so on. The variables $x$

and $y$ can represent any uncertain parameter, such as the measurement uncertainty or the production rate of the nuclide. All uncertainties (e.g., nuclide concentration) are assumed to be at the one sigma level unless otherwise stated. The derivatives in Eq. (17) are calculated using the nominal value plus

the associated uncertainty and then recalculating the denudation rate in the original, pixel-by-pixel fashion.

Three uncertainties are included in the calculation: i) the uncertainty in cosmogenic nuclide concentration, ii) the uncertainty in the production rate at sea level, high latitude ($P_{i,SLHL}$), and iii) uncertainty in muon production. Uncertainty in cosmogenic nuclide concentration is reported by authors alongside concentrations. For the cosmogenic nuclide concentration uncertainty, the concentration is used directly to determine the denudation rate uncertainty. For all other parameters, the

uncertainty values help to predict a new concentration in each pixel, which is then used to determine denudation rate uncertainty. It is important to note here that we do not calculate uncertainties inherent in the basin-averaging approach which assumes spatial homogeneity in source material and denudation rates, and denudation that is steady in time; we address these uncertainties in Sect. 5.4.

    The uncertainty on the production rate ($P_{i,SLHL}$) is based on that used in the CRONUS-2.2 cal-

culator (Balco et al., 2008): in CRONUS-2.2 the uncertainty is 0.39 atoms cm$^{-2}$ yr$^{-1}$ for $^{10}$Be based on a production rate of 4.49 atoms cm$^{-2}$ yr$^{-1}$. This means the uncertainty in CRONUS-2.2 is 8.7% of $P_{i,SLHL}$ for $^{10}$Be. We use this uncertainty for both $^{10}$Be and $^{26}$Al based on our the production rates reported in Table (2). Although the recent CRONUS-Earth calibration (Borchers et al., 2016) has produced new production rates for both $^{10}$Be and $^{26}$Al, the production rate uncertainties remain

in the same range as those used here (Phillips et al., 2016a).

    Field studies have shown that muon production based on laboratory experiments (Heisinger et al., 2002a, b) overestimate muon production observed in deep samples (Braucher et al., 2003, 2011, 2013; Balco et al., 2013; Phillips et al., 2016a), there is still some uncertainty over the exact muon production profile. CAIRN employs the exponential scaling method from Braucher et al. (2009). It

then calculates the upper bound of uncertainty derived uncertainty in muon models by calculating the difference between the default CAIRN muon model and those from the Schaller et al. (2009) scheme, which approximates the original Heisinger results (Heisinger et al., 2002a, b).

### 4.2   Uncertainty from snow shielding

Uncertainties from nuclide concentration, muon production, and production rates are calculated in-

ternally by our software. Uncertainties from snow and self shielding rely on user-supplied information and therefore must be estimated separately.

    Snow shielding can be supplied as a constant effective snow thickness (in g cm$^{-2}$) or spatially distributed information in the form of a raster. Most snow shielding calculations reported in the literature are based on an effective attenuation estimated by the thickness of snow (e.g., Balco et al.,

2008), but recent field-based measurement indicate that snow may attenuate fluxes of cosmic rays to a greater extent than assumed in simple mass-based snow shielding calculations (Zweck et al., 2013; Delunel et al., 2014). However these uncertainties are small compared to the extreme uncertainties of the thickness, extent and duration of snow over millennial timescales, which are unlikely to ever

be well constrained. If no snow shielding values are provided, the software assumes that there is no

snow cover.

To calculate uncertainties, users must supply two scenarios for these shielding factors. For example, the user could provide two snow thickness rasters representing variation in snow thickness with $1\sigma$ uncertainty (how an author might calculate this could fill another paper and is beyond the scope of our study). The denudation rates of these two scenarios would then be calculated, and the square

of the difference in these two denudation rates would then be inserted into Eq. (17). In this way users can calculate shielding uncertainties manually.

### 4.3    Summary of CAIRN parameters for denudation calculations

To summarize, CAIRN predicts cosmogenic nuclide production from neutrons and muons using a four exponential approximation of data from Braucher et al. (2009). These production rates are

scaled using Lal/Stone time-independent scaling. Production is calculated at every pixel, with atmospheric pressure calculated via interpolation from the NCEP2 reanalysis data (Compo et al., 2011). Topographic shielding is calculated using the method of Codilean (2006), and scaled production rates are multiplied by topographic, snow, and self shielding at each pixel. Decay rates, attenuation lengths, and parameters for production are reported in Table 2. Denudation rates are reported in g

$cm^{-2}$ $yr^{-1}$ because in these units no assumptions about density, which is spatially heterogeneous, are required. In addition, users must report the AMS standard when supplying nuclide concentrations to CAIRN and the concentrations are then normalized following the same scheme as Balco et al. (2008). The CAIRN software prints these parameters to a file so that if they change in the future based on new calibration datasets, users will be able to both view and report these updated

values.

### 5    Spatial averaging for ingestion by other denudation rate calculators

In addition to producing denudation rates, CAIRN also provides spatially-averaged production rates and effective catchment-averaged pressure (see Sect. 5.3), so that users can compute denudation rates using other available calculators. Programs such as the CRONUS-Earth calculators, referred

to as CRONUS-2.2 for Balco et al. (2008) and CRONUScalc for Marrero et al. (2016), and COS-MOCALC do not have the ability to calculate catchment-averaged parameters. CAIRN can be used independently to determine production rates or in conjunction with these other calculators, which allows for the possibility of using time-dependent scaling and other new features in the future.

### 5.1    Conversion of depth integrated parameters for calculator ingestion

CAIRN iterates on denudation rate until the predicted cosmogenic concentrations from Eq. (14) is reached. Eq. (14) is a depth integrated approach that is a direct solution of the production equations.

This depth-integrated solution subsumes both snow and self shielding. This is different from from COSMOCALC and the CRONUS calculators, which take separate values for shielding. Thus to pass results from CAIRN to calculators we must first calculate equivalent snow and self shielding values for each pixel. Note that these values are not used within denudation rate calculation in CAIRN, they are only used when shielding values are passed to the COSMOCALC and the CRONUS calculators.

Self shielding used for spatial averaging is calculated for each pixel $k$ with:

$$S_{self,k} = \frac{\Lambda_{i,0}}{d_{t,k}} \left( 1 - e^{-\frac{d_{t,k}}{\Lambda_{i,0}}} \right) \tag{18}$$

where $S_{self,k}$ is the self shielding correction for the $k^{th}$ pixel, $d_{t,k}$ is the shielding thickness for the $k^{th}$ pixel (in g cm$^{-2}$). Equation (18) is used in both COSMOCALC and CRONUS. In the CRONUS calculators, snow shielding is lumped with topographic shielding, therefore the CRONUS calculators presume the user will determine the product of snow and topographic shielding at a site with a method of their choice. COSMOCALC includes a snow shielding calculator which assumes that the equivalent depth of snow (in g cm$^{-2}$) attenuates neutron production following the formula:

$$S_{snow,k} = e^{-\frac{d_{s,k}}{\Lambda_{i,0}}} \tag{19}$$

where $S_{snow,k}$ is the snow shielding correction of the $k^{th}$ pixel and $d_{s,k}$ is the time-averaged depth of snow water equivalent in g cm$^{-2}$. We adopt this approximation when performing spatial averaging. Recent work suggests snow may attenuate spallation to a greater degree than predicted by Eq. (19) (Delunel et al., 2014), and Zweck et al. (2013) suggest that the attenuation length for snow is reduced compared to rock (they report an attenuation length of 109 g cm$^{-2}$ for snow). However, the uncertainty in historic snow thickness vastly outweighs uncertainties from the snow shielding equation. Although there have been methods suggested to model the evolution of snow thickness through time (e.g., Beniston et al., 2003), the averaging time for eroded particles that accumulate cosmogenic nuclides is on the order of thousands to tens of thousands of years (e.g., Lal, 1991), and reconstructing snow thickness over this timescale is highly uncertain. Users wishing to approximate the Zweck et al. (2013) attenuation lengths can feed CAIRN snow rasters with a thicker apparent snow layer. Overall, we therefore recommend that users include a large range of snow thickness in their uncertainty analysis, guided by historical observations of snow depth.

## 5.2  Spatial averaging for COSMOCALC

In COSMOCALC's erosion calculator (which calculates denudation), the required inputs are a combined shielding and scaling term, the cosmogenic nuclide concentration and the uncertainty in the cosmogenic nuclide concentration. That is, scaling and shielding are combined in a single, spatially

averaged term. We calculate the scaling factor $S_{CCtot}$, which is a lumped shielding and scaling term, with

$$S_{CCtot} = \frac{1}{N}\sum_{k=0}^{N} S_{snow,k}S_{topo,k}S_{self,k}S_{i,k} \qquad (20)$$

where terms are calculated on a pixel-by-pixel basis. Snow shielding is calculated from Eq. (19), self shielding is calculated from Eq. (18), and topographic shielding is calculated accounting for the effects of sloping samples and topography blocking cosmic rays (see Sect. 2.4). We wish to emphasize that CAIRN reports $S_{CCtot}$ for users that wish to use it in COSMOCALC, whereas the denudation rates reported by CAIRN use Eq. 14 for snow and self shielding. Production scaling for cosmogenic nuclide $i$ at pixel $k$, $S_{i,k}$, is calculated using Eq. (16) and Lal/Stone scaling (Sect. 2.5).

### 5.3 Spatial averaging for the CRONUS calculators

The CRONUS calculators (CRONUS-2.2 and CRONUScalc) require a lumped shielding value and information about either the elevation or pressure of the sample. Spatial averaging of the lumped shielding value, $S_{CRshield}$, is calculated with:

$$S_{CRshield} = \frac{1}{N}\sum_{k=0}^{N} S_{snow,k}S_{topo,k}S_{self,k} \qquad (21)$$

Note that we fold the self shielding into the lumped shielding term so that when transferring data to the CRONUS calculator the sample thickness should be set to 0.

The CRONUS calculators then calculate production using either an elevation or pressure. Production rates are nonlinear with either elevation or pressure, so we must compute an effective pressure that reproduces the mean production rate in the catchment. This is because the arithmetic average of either elevations or pressures within the catchment, when converted to production rate, will not result in the average production rate due to this nonlinearity. CAIRN calculates an effective pressure that reproduces the effective production rate over the catchment. The average production rate is calculated with:

$$S_{effp} = \frac{1}{N}\sum_{k=0}^{N} S_{i,k} \qquad (22)$$

We then use the Newton iteration on the Lal/Stone scaling scheme to find the pressure which reproduces the basin average production rate ($S_{effp}$). That way, results from our method can be compared to results from the CRONUS calculator and, if users are so inclined, they can use time varying production scalings via the CRONUS calculator (which CAIRN does not include for reasons outlined in Sect. 2.5).

### 5.4 Uncertainties introduced by spatial and temporal variability

CAIRN provides uncertainty estimates based on uncertainties in the measurement of nuclide concentrations, and uncertainties in production rates. It does, however, make an assumption of steady erosion, and also makes assumptions likely to be violated almost everywhere on Earth due to the long timescales of geomorphic adjustment, which are on the order of tens of thousands to millions of years (e.g., Fernandes and Dietrich, 1997; Roering et al., 2001; Whipple, 2001; Mudd and Furbish, 2007; Braun et al., 2015) versus climate oscillations that are tens to hundreds of thousands of years (e.g., Lisiecki and Raymo, 2005). In addition, spatial heterogeneity in lithology and target mineral concentrations can lead to additional uncertainty to denudation rate estimates (e.g., Safran et al., 2006; Carretier et al., 2015). Mass wasting can also perturb the concentration of cosmogenic nuclides (e.g., Niemi et al., 2005; Yanites et al., 2009), leading to further uncertainties. Finally, as noted in Sect. 4.2, if snow shielding is to be taken into account, one must estimate the shielding provided by snow over millennial timescales, which, to put it mildly, are difficult to constrain.

For the problem of spatially heterogeneous lithology, careful geologic mapping, such as that done by Safran et al. (2006) and Carretier et al. (2015) can alleviate some of the uncertainty, but such mapping is logistically challenging. For landsliding, mass removal can be measured in the field, modelled (e.g., Niemi et al., 2005; Yanites et al., 2009), or approximated using mapped landslide inventories (e.g., Hovius et al., 1997; Korup, 2005). These may be combined with data on landslide area-volume relationships (e.g., Guzzetti et al., 2009). The main difficulty here is that it takes some time for the cosmogenic nuclide concentration to readjust after mass removal (e.g., Schaller and Ehlers, 2006; Muzikar, 2009; Mudd, 2016) and thus one must make some estimate of not only the spatial distribution of landslides but their evolution through time (Yanites et al., 2009). Simulating nuclide concentrations in settings where denudation rates vary in space and time is possible (Mudd, 2016), but computationally intensive and one must have some confidence that one can accurately reconstruct the temporal evolution of denudation rates. Although recent progress has been made in deriving time series of denudation rates from current topography (e.g., Whittaker et al., 2008; Pritchard et al., 2009; Hurst et al., 2013; Goren et al., 2014; Fox et al., 2014; Croissant and Braun, 2014; Rudge et al., 2015), these methods still suffer from the fact that we lack devices for time travel and struggle to test such reconstructions.

Ultimately, uncertainties in the spatial distribution of denucation and source material, and temporal uncertainties in denudation rates, mean that the uncertainties reported by CAIRN are the minimum uncertainties: they do not take into account landscape transience, lithology, or variation in snow shielding. The fact that catchment-averaged denudation rates carry additional uncertainties is well known, and Dunai (2010) estimates that any catchment-averaged denudation rate carries with it a minimum 30% uncertainty. Because the uncertainties mentioned in this section are difficult, if not impossible to constrain, our approach with CAIRN is to report the uncertainties that can be con-

strained and caution users that there are large additional unconstrained uncertainties related to the assumptions underpinning the method.

## 6  Method comparison

Comparison with other methods is difficult because authors reporting cosmogenic nuclide-derived catchment-averaged denudation rates have not made their algorithms available as open-source tools. Our spatially-averaged production scaling and shielding estimates are approximations of spatial averaging reported by other authors. We compare our data to both published denudation rate estimates, and to estimates of denudation rates generated by the CRONUS calculator given the spatial averaging described in Section 5.3. In our comparisons we use seven published cosmogenic datasets (Table 3). These datasets were chosen to span a wide range of locations (i.e., differing latitudes and elevations) and denudation rates. The parameters used by CAIRN for these comparisons are reported in Table 2.

It will perhaps aid the reader if we explain how denudation rate estimates may vary between methods. Firstly, production rates are nonlinearly related to elevation, and thus spatial averaging of the product of production scaling and shielding is not the same as the product of the spatial averages of production scaling and shielding. In addition, previous studies and other calculators have chosen different parameters for cosmogenic nuclide production and shielding. For example, past publications have used a wide variety of methods for estimating topographic shielding (e.g., see Table 3). Choices of spallation and muon production rates also affect the final denudation rate. Consider a measured nuclide concentration that one uses to infer a denudation rate. If one assumes a high production rate (via either muons or spallation), it means that for a given denudation rate the predicted nuclide concentration is higher. Thus, for a given nuclide concentration, the inferred denudation rate is higher if the assumed production rate is higher (see dashed lines in Figure 1). If the inferred shielding is higher, then for a given denudation rate the production is lower, and the inferred denudation for a given concentration will be lower.

### 6.1  Spatial averaging of production and shielding vs pixel-by-pixel calculations

First, we compare results of two methods using the exponential approximation of muon production (Eq. 12), used in both COSMOCALC and the CAIRN calculator. The difference in calculating denudation rates by iterating upon cosmogenic nuclide concentration from all pixels in a basin (the CAIRN method) and calculating it by using a spatial average of the production of scaling and production terms (Eq. 20) is virtually zero if snow and self shielding are spatially homogenous (Figure 2a). Thus we find that combining all scaling and shielding terms in a single lumped term is adequate for calculating denudation rates if computational power is limited.

Separating production rate scaling from shielding leads to slightly larger uncertainty (Figure 2b), but in terms of the total uncertainty this averaging also leads to small uncertainties (on the order of 1-2% compared to 10-20% from other sources of uncertainty). We suspect that many users will want to compare rates determined by our software with the popular CRONUS calculators (Balco et al., 2008; Marrero et al., 2016). The CRONUS calculators internally scale production rates while shielding is supplied by the user. Consequently, the uncertainties plotted in Figure 2b approximate uncertainties arising from the spatial averaging process that users must pass to the CRONUS calculators. Some users may wish to calculate denudation rates using time-dependent scaling schemes, which is not possible in CAIRN, but CAIRN can be used as a front end to the CRONUS calculators via its spatial averaging capabilities with the confidence that this will only introduce relatively small errors.

### 6.2   Comparison with existing denudation rate estimates

Denudation rates reported in the literature from catchment-averaged cosmogenic nuclide concentrations are calculated using a wide variety of methods. The term erosion rate is often substituted for denudation rate although few studies attempt to account for chemical weathering (cf., Kirchner et al., 2001; Riebe et al., 2001). Studies differ in their strategies for production rate scaling, topographic, snow, and self shielding, and the manner in which spatial averaging is performed. In many cases there is insufficient detail reported that might enable other groups to reproduce reported denudation rates. A primary motivation behind CAIRN is to provide an open-source means of computing denudation rates that may then be reproduced by other groups. We have incorporated reported snow shielding from previous studies by inverting Eq. 19 for an annual average snow thickness and then distributing this thickness over the entire DEM. We acknowledge this is a poor representation of snow thickness but snow shielding rasters are rarely available and in most cases there is little reported snow shielding.

The diversity in methods for calculating denudation rates reported in the literature means that it is difficult to compare denudation rates when they come from different studies. This problem has been highlighted by previous data intercomparison studies (Portenga and Bierman, 2011; Willenbring et al., 2013a; Harel et al., 2016). High latitude production rates under Lal/Stone scaling of $^{10}$Be have changed in the last 10 years due to an ever increasing number of calibration sites (e.g., Phillips et al., 2016a) and changing AMS standards (Nishiizumi, 2004). In some cases, muons are not considered, whereas other studies use a variety of different muon production schemes (e.g., Table 3). Topographic shielding is occasionally not considered (particularly in older studies). In some cases the horizon elevation is recorded from a limited number of directions (e.g., COSMOCALC includes a calculator using 8 directions), and in other instances the computational method of Codilean (2006) is used. Studies also cite Dunne et al. (1999) for shielding but this paper lists several methods for calculating shielding: the equations therein depend on the number and geometry of shielding objects

and this information is seldom reported. Even when the more robust method of Codilean (2006) is used, the spacing of azimuth and angle of elevation is often not reported.

Studies typically report erosion or denudation rates in dimensions of length per time, but this requires an assumption about density, which can vary spatially and is sometimes not reported. Most studies use a rock equivalent denudation rate (as opposed to a regolith or soil denudation rate) and thus densities assumed are typically rock densities (see Table 3). Because denudation rates are traditionally reported in dimensions of length per time, we do not suggest future authors cease reporting denudation in these dimensions, but we do recommend also reporting denudation rates in dimensions of mass per area per time (e.g., $g\ cm^{-2}\ yr^{-1}$) because these units allow simpler comparison between sites as they require no assumptions about spatially heterogeneous density.

Of our seven example datasets (Table 3), only 3 of the original authors reported topographic shielding factors. We calculated shielding using the CAIRN method with $\Delta\phi = 5°$, $\Delta\theta = 8°$ in these three high relief landscapes using a 90 meter resolution DEM. Our small values of $\Delta\phi$ and $\Delta\theta$ lead to variations in shielding between CAIRN and reported values (Figure 3). Authors typically do not give enough information to reproduce their shielding calculations, but we note that authors that employ the equations of Dunne et al. (1999) use a limited number of horizon measurements to calculate shielding. For example in COSMOCALC (Vermeesch, 2007), users are expected to input horizon values at $45°$ intervals. Our calculations suggest that this can lead to lower maximum shielding differences between this method and the CAIRN method (Table 1). An example of the potential underestimates of topographic shielding is shown in Figure 4.

The denudation rates predicted by CAIRN are plotted against reported denudation rates in Figure 5. These data are scattered about the 1:1 line, but for most samples the CAIRN denudation rate is lower than the reported denudation rate. Reasons for this vary since the method used to calculate denudation rates vary in each example study, but differences are likely to be due to the higher production rates used in previous studies (Table 3) and slightly greater topographic shielding in CAIRN (see Figure 1).

One component of CAIRN that requires caution is that the snapping of cosmogenic samples to channels is automated: if errors in the DEM place the main channel in the wrong location, or GPS coordinates of the sampling location contain large errors (common in older datasets), there is a chance the basin selected by CAIRN will not be the same as the sampled basin. This can result in large errors as production rates vary significantly with elevation. We have provided a tool in the github repository that allows users to check the basins that are associated with cosmogenic nuclide samples. If these do not match the expected basins, then users will need to manually change the latitude and longitude of the samples until they are located near the correct channel.

We wish to emphasize that the relative denudation rates do not change significantly between CAIRN and reported values (as evidenced by a clustering about the 1:1 line in Figure 5). In addition previous studies contain elements modulating denudation rates that are not contained within

the current version of CAIRN. For example, Kirchner et al. (2001) reports true physical erosion rather than denudation and Safran et al. (2006) modified their denudation rates based on the quartz content of the source areas.

### 6.3 Comparison with the CRONUS calculators

The results from CAIRN are compared to results from both CRONUS calculators. When comparing output from CAIRN with output from the online CRONUS-2.2 calculator, far larger uncertainties (up to to 40% of the denudation rate) occur. These differences are not controlled by denudation rate (Figure 6a) but are instead mainly a function of the production rate (Figure 6b). In the previous section, we found that differences due to spatial averaging and separation of shielding from production scaling are small. The large difference is primarily due to the difference in spallation production rates and the over-production of muons in CRONUS version 2.2, as described by Balco et al. (2013). According to Balco et al. (2013), future versions of this CRONUS calculator will be updated to have significantly reduced muogenic production consistent with recent studies (Braucher et al., 2003, 2011, 2013; Phillips et al., 2016a). If production rates in CRONUS are changed to reflect the production rates from Braucher et al. (2011), we find that differences are quite small (Figure 7). We see from this figure that in locations with high production rates just under half of these differences between CAIRN and CRONUS-2.2 are from the different spallation rates, whereas in locations with low production rates, most of the differences are due to the higher muon production present in CRONUS-2.2.

The other CRONUS calculator, CRONUScalc, incorporates new spallation production rates and muon production is calculated using production rates based on a deep core from Antarctica (Marrero et al., 2016; Phillips et al., 2016a). In order to examine the underlying source of discrepancies between the three calculators, we plot the total and muon production rates for the CAIRN, CRONUS-2.2, and CRONUScalc calculators in Figure (8). The production rates for CRONUS-2.2 are calculated directly from the MATLAB scripts available online. The CRONUScalc production rates are approximated as a three exponential analytical function with parameters shown in Table 4. Although total production rates appear relatively similar, CRONUScalc and CAIRN predict significantly smaller muon contributions that CRONUS-2.2. The result is that for the same denudation rate, the CRONUS-2.2 calculator produces significantly more (in some cases 40% more) atoms than using CAIRN or CRONUScalc (Figure 9). This discrepancy between muon production is important because rapidly eroding samples accumulate a significant proportion of their nuclide concentrations below 100 g cm$^{-2}$, leading to a large discrepancy in calculated denudation rates between CRONUS-2.2 and the other two calculators (CAIRN and CRONUScalc), which both incorporate more recent muon models. The CAIRN outputs of topographic shielding, as well as the spatial averaging of both production scaling and shielding, are independent of these calculators and will still provide spa-

tial averaging for use with future calculator versions, even as production rates and mechanisms are updated.

We have used the spatially averaged shielding and scaling outputs from CAIRN to determine differences between CAIRN and CRONUScalc. We find that there is a 2.5% to 5% difference between the denudation rates predicted by CAIRN and those predicted by CRONUScalc (Figure 10). Currently CRONUScalc is not able to calculate very high denudation rates (for rates greater than ~0.06 g cm$^{-2}$ yr$^{-1}$ the current version of CRONUScalc crashes; it was designed for exposure ages and be-

comes computationally unstable at high erosion rates) so we cannot compare CAIRN to CRONUS-calc for all of the example datasets. The differences in Figure (10) arise from two sources: first, we must pass the product of the scaling ($S_{effp}$) and shielding ($S_{CRshield}$) to CRONUScalc rather than calculating pixel by pixel values. Second, the default muon production in CAIRN is derived from the Braucher et al. (2009) scheme, which is slightly different than the production schemes derived

from Marrero et al. (2016) and Phillips et al. (2016a) (see Figure 8). In CAIRN, users can choose the muon production scheme, and we have implemented an approximation of the muon production scheme from Marrero et al. (2016) that uses the exponential form of Eq. 2 (see Table 4). It is important to note that the CAIRN implementation of muons from Marrero et al. (2016) assumes that $\Lambda = 160$ g cm$^{-2}$ for spallation, whereas in CRONUScalc this attenuation length can vary as a func-

tion of latitude and pressure. We compare the denudation rates from CAIRN using the production parameters in Table 4 ($\epsilon_{CAIRN-CRC}$) with the default production scheme of Braucher et al. (2009) in Figure (11). The differences here are smaller (mostly less than 2%) suggesting that much of the difference seen in Figure (10) is due to spatial averaging.

## 7    Conclusions

We present an automated, open-source method for calculating catchment-averaged denudation rates based on the concentrations of *in-situ* cosmogenic nuclides collected in stream sediment. Our catchment-averaged denudation rate method (CAIRN) predicts cosmogenic nuclide concentrations based on pixel-by-pixel scaling and shielding. These concentrations are then averaged to predict the catchment-averaged concentration. Newton iteration is then used to find the denudation rate for which the pre-

dicted concentration matches the measured concentration and to derive associated uncertainties. In addition, CAIRN provides spatially averaged shielding and scaling values that can be used by other popular calculators (which do not provide spatial averaging, e.g., CRONUS and COSMOCALC). The CAIRN method is provided as open-source software so that reported denudation rates can be easily reproduced.

The CAIRN method is intended to streamline the computation and reporting of catchment-averaged denudation rates, but it has limitations that may be the subject of future developments. At the moment CAIRN assumes steady erosion; there is no facility for incorporating transient erosion rates

which might affect nuclide concentrations in transient landscapes (e.g., Willenbring et al., 2013b; Mudd, 2016). In addition, the method does not include a facility for nesting basins in which the denudation rate in a large basin incorporates the denudation rates from smaller basins that it contains. The calculator cannot account for differing source areas of material, so at the moment it is not capable of using different particle size fractions to identify denudation hot spots (e.g., Riebe et al., 2015; Carretier et al., 2016). Despite these limitations, the CAIRN method addresses the need to provide transparent, reproducible estimates of denudation rates.

Our open source framework allows other users to update the algorithms (e.g., a nesting function could be built on top of the current CAIRN architecture) and different atmospheric reanalysis data or new muon scaling schemes can be added as needed in the future. Thus we hope it will provide a platform for more nuanced estimates of denudation rates from cosmogenic nuclides in the future.

**Software and data availability**

The software is available at the LSDTopoTools Github website (https://github.com/LSDtopotools/). Instructions for installing the software and its use are located withing the LSDTopoTools documentation website (http://lsdtopotools.github.io/LSDTT_book/). The data files containing formatted cosmogenic data, parameter values and results, and scripts for plotting figures used in this paper are also located on the Github site. All DEMs used in the analysis were derived from Shuttle Radar Topography Mission 3 arc second data available from the United States Geological Survey digital globe website (http://earthexplorer.usgs.gov/).

**Author contributions**

S.M. Mudd (SMM), MDH and SWDG wrote the software. MAH, SMM and S.M. Marrero analyzed the data. SMM wrote the paper with contributions from other authors.

*Acknowledgements.* SMM and MAH are funded by U.S. Army Research Office contract number W911NF-13-1-0478 and SMM and SWDG are funded by NERC grant NE/J009970/1. S.M. Marrero is funded by NERC grant NE/I025840/1. This paper is published with the permission of the Executive Director of the British Geological Survey (NERC), and was supported by the Climate and Landscape Change research programme at the BGS. We would like to thank Associate Editor Josh West for his helpful comments and also for testing the code. We would also like to thank an anonymous reviewer and Greg Balco for their constructive and beneficial comments which significantly improved the paper.

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

**Table 1.** Absolute maximum residuals (i.e., greatest residual within the DEM) for different combinations of $\Delta\theta$ and $\Delta\phi$ used in shielding calculations for a high relief basin in the Himalayas.

| $\Delta\phi$, degrees | $\Delta\theta$, degrees | | | | | | | | | |
|---|---|---|---|---|---|---|---|---|---|---|
| | 1 | 2 | 3 | 5 | 8 | 10 | 15 | 30 | 45 | 60 |
| **1** | 0.000 | 0.002 | 0.004 | 0.009 | 0.010 | 0.011 | 0.027 | 0.053 | 0.063 | 0.081 |
| **2** | 0.004 | 0.004 | 0.005 | 0.009 | 0.010 | 0.012 | 0.029 | 0.057 | 0.064 | 0.080 |
| **3** | 0.008 | 0.008 | 0.008 | 0.010 | 0.011 | 0.012 | 0.027 | 0.053 | 0.062 | 0.081 |
| **5** | 0.014 | 0.015 | 0.016 | 0.017 | 0.018 | 0.018 | 0.030 | 0.056 | 0.065 | 0.087 |
| **8** | 0.023 | 0.023 | 0.026 | 0.025 | 0.027 | 0.030 | 0.039 | 0.064 | 0.082 | 0.093 |
| **10** | 0.036 | 0.037 | 0.033 | 0.040 | 0.035 | 0.040 | 0.037 | 0.063 | 0.074 | 0.104 |
| **15** | 0.057 | 0.059 | 0.058 | 0.060 | 0.060 | 0.058 | 0.065 | 0.084 | 0.100 | 0.122 |
| **20** | 0.072 | 0.071 | 0.073 | 0.075 | 0.077 | 0.076 | 0.083 | 0.111 | 0.109 | 0.138 |
| **30** | 0.171 | 0.172 | 0.168 | 0.176 | 0.167 | 0.167 | 0.173 | 0.188 | 0.160 | 0.242 |
| **45** | 0.337 | 0.340 | 0.332 | 0.335 | 0.346 | 0.335 | 0.332 | 0.393 | 0.385 | 0.430 |
| **60** | 0.352 | 0.352 | 0.352 | 0.352 | 0.352 | 0.352 | 0.352 | 0.352 | 0.385 | 0.418 |

**Table 2.** Default parameters used in the CAIRN model.

| Parameter | Value | Source |
|---|---|---|
| $\lambda_{10Be}$ | $500 * 10^{-9}$ yr$^{-1}$ | Chmeleff et al. (2010); Korschinek et al. (2010) |
| $\lambda_{26Al}$ | $980 * 10^{-9}$ yr$^{-1}$ | Nishiizumi (2004) |
| $\Lambda_i$ | 160;1500;4320 g cm$^{-2}$ | From COSMOCALC version 2.0 to mimic Braucher et al. (2011) |
| $^{10}$Be $P_{SLHL}$ | 4.30 atoms g$^{-1}$ yr$^{-1}$ | From COSMOCALC version 2.0 to mimic Braucher et al. (2011) |
| $^{10}$Be $F_i$ | 0.9887; 0.0027; 0.0086 (dimensionless) | From COSMOCALC version 2.0 to mimic Braucher et al. (2011) |
| $^{26}$Al $P_{SLHL}$ | 31.10 atoms g$^{-1}$ yr$^{-1}$ | From COSMOCALC version 2.0 |
| $^{26}$Al $F_i$ | 0.9699; 0.00275; 0.0026 (dimensionless) | From COSMOCALC version 2.0 to mimic Braucher et al. (2011) |

**Table 3.** Datasets used for method comparisons. $^{10}$Be production rate (Prod rate) is given for sea level, high latitude and in units of atoms g$^{-1}$ yr$^{-1}$. 'CR' or 'CR muons' refers to the spallation or muon calculation methods and production rates used in CRONUS-2.2 (Balco et al., 2008). The scaling values, production rates, topographic shielding and notes reported in this table are for the original studies: CAIRN uses the same settings (see Table (2) for its calculations regardless of site location.

| Study | Location | Scaling | Prod rate | Topo Shielding | Other Notes |
|---|---|---|---|---|---|
| Bierman et al. (2005) | New Mexico, USA | Lal/Stone | 5.2 | None. | $\rho = 2.7$ g cm$^{-3}$, no muons. |
| Dethier et al. (2014) | Colorado, USA | Lal/Stone | 4.49 (CR) | None. | $\rho = 2.7$ g cm$^{-3}$, fast muons only. |
| Kirchner et al. (2001) | Idaho, USA | Lal/Stone | 4.72 | Dunne et al. (1999), details not given. | Corrections for chemical weathering. |
| Munack et al. (2014) | Ladakh, India | Lal magnetic | 4.49 (CR) | Pixel-by-pixel, but details not given. | CR muons. Snow and ice shielding considered. |
| Palumbo et al. (2010) and Palumbo et al. (2011) | Tibet | Dunai (2000) | 5.12 | Codilean (2006), $\Delta\phi$, $\Delta\theta$ not reported. | Muons using Granger and Smith (2000) scheme. $\rho = 2.65$ g cm$^{-3}$. |
| Safran et al. (2006) | Bolivia | Dunai (2000) | None. | No muons. $\rho$ not reported. Corrections for quartz fraction. | |
| Scherler et al. (2014) | Garwahl Himalaya | Lal magnetic | 4.49 (CR) | Pixel-by-pixel, but details not given. | CR muons. Snow and ice shielding considered. |

**Table 4.** Parameters used for production of $^{10}$Be which approximate the scheme in CRONUScalc (Marrero et al., 2016). $\lambda_{10Be}$ values are the same as defaults listed previously. The $F_i$ values represent spallation and fast and slow muons, respectively.

| Parameter | Value |
|---|---|
| $\Lambda_i$ | 160;1460;11040 g cm$^{-2}$ |
| $^{10}$Be $P_{SLHL}$ | 4.075 atoms g$^{-1}$ yr$^{-1}$ |
| $^{10}$Be $F_i$ | 0.9837; 0.0137; 0.0025 (dimensionless.) |

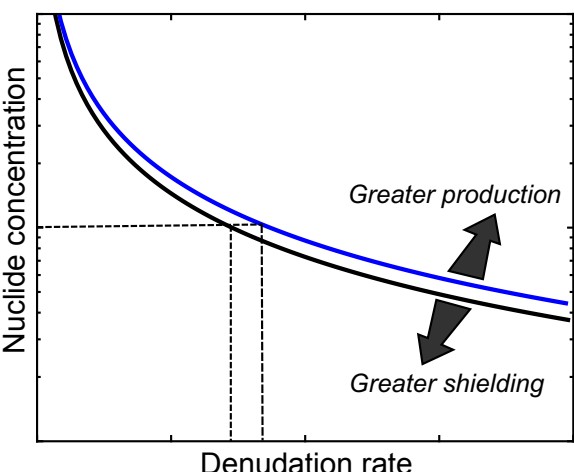

**Figure 1.** A schematic drawing of the predicted concentration of a nuclide as a function of denudation rate. If production rates are assumed to be higher, the predicted concentration will be higher for a given denudation rate. If shielding is greater, the predicted concentration is lower for a predicted denudation rate. Thus assumptions about production and shielding will affect the inferred denudation rate given a sample with fixed concentration, shown with the dashed lines.

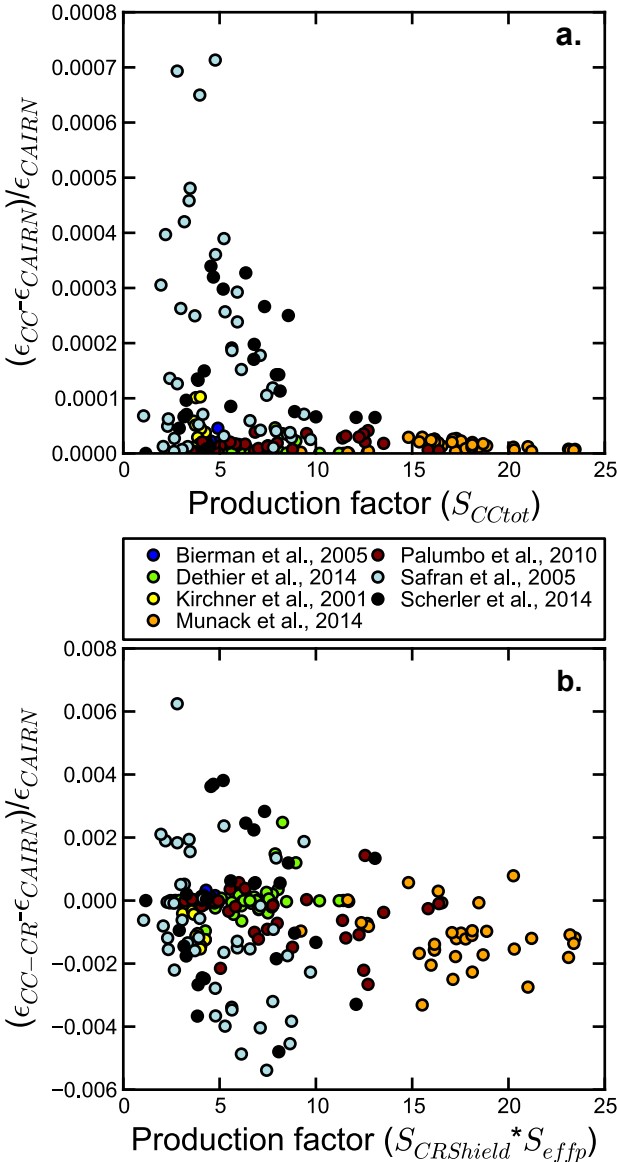

**Figure 2.** Differences between the denudation rate calculated by CAIRN ($\epsilon_{CAIRN}$) and the denudation rate using the production factor ($S_{CCtot}$) (which includes production scaling and shielding) passed to COSMOCALC ($\epsilon_{CC}$) (**a.**), and differences between the denudation rate calculated by CAIRN ($\epsilon_{CAIRN}$) and the denudation rate using separate spatial averages for shielding and production scaling that are then averaged ($\epsilon_{CC-CRONUS}$) as a function of production factor (**b.**). In this case the production factor is calculated by multiplying the separately averaged shielding ($S_{CRShield}$) and scaling ($S_{effp}$) factors. This approach emulates the data requirements for CRONUS-2.2, which calculates production scaling and accepts a single shielding factor (for snow and topography combined). Although the shielding and scaling emulate data requirements for CRONUS-2.2, the denudation rate is calculated using the exponential production method of CAIRN and COSMOCALC.

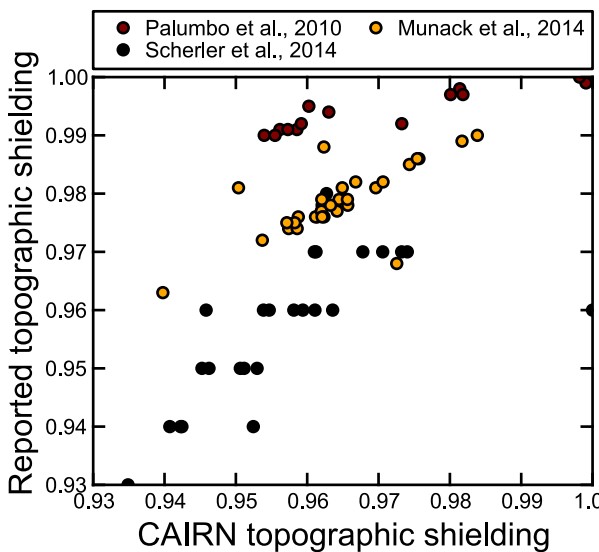

**Figure 3.** Topographic shielding ($S_t$) calculated using $\Delta\phi = 5°$, $\Delta\theta = 8°$ plotted as a function of reported shielding.

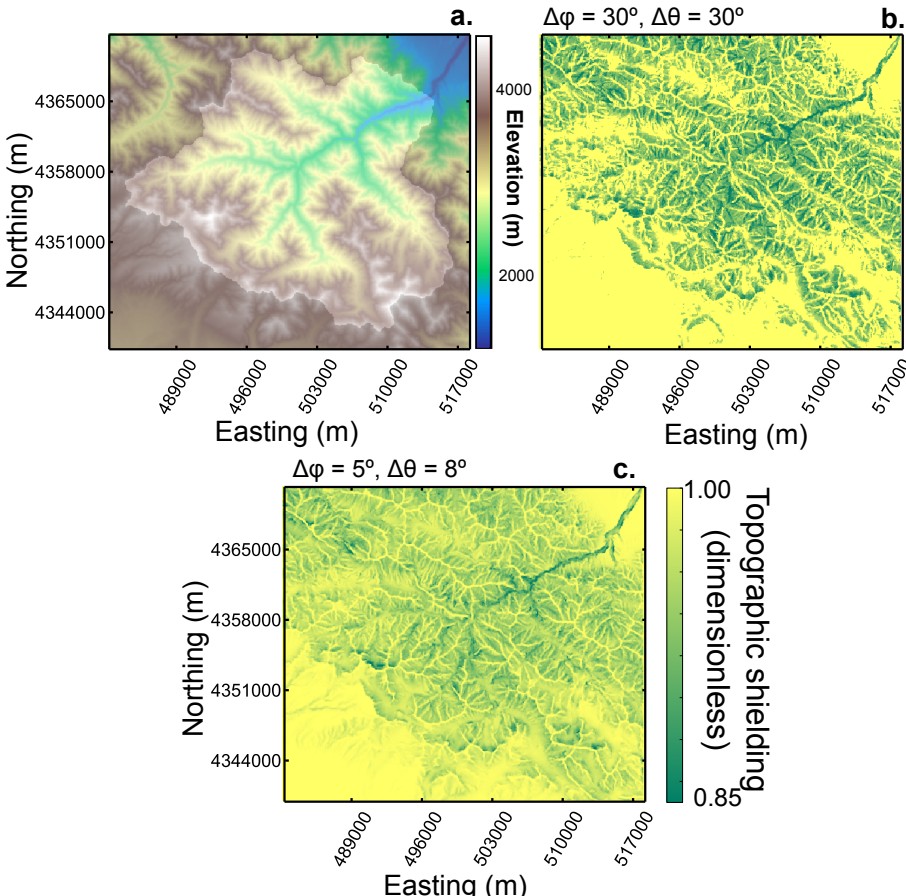

**Figure 4.** Comparison of the topographic shielding for different values of $\Delta\phi$ and $\Delta\theta$. The Tibetan basin is for sample 07C13 in Palumbo et al. (2011). Maps are projected into WGS1984, UTM zone 47N. The basin is shown in plot **(a)**, whereas the topographic shielding factor is shown in plots **(b)** and **(c)**.

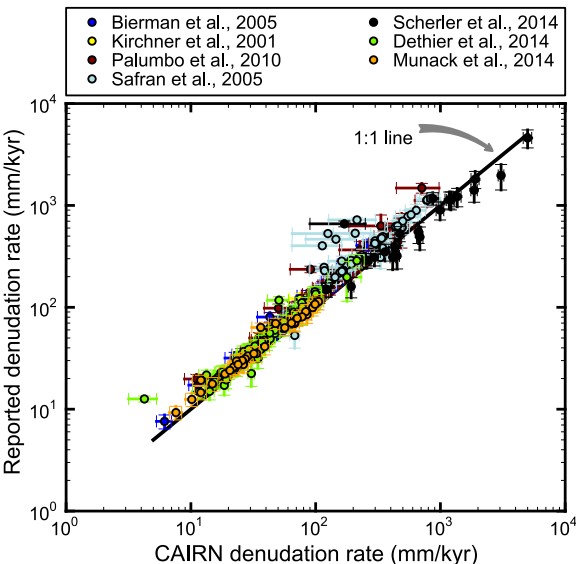

**Figure 5.** Comparison of denudation rates reported by selected studies plotted against denudation rates predicted by CAIRN. The denudation rates for individual studies use their original assumptions of the density of the surface material, as reported in Table 3. The results from CAIRN in this plot use a density of 2.65 g cm$^{-2}$.

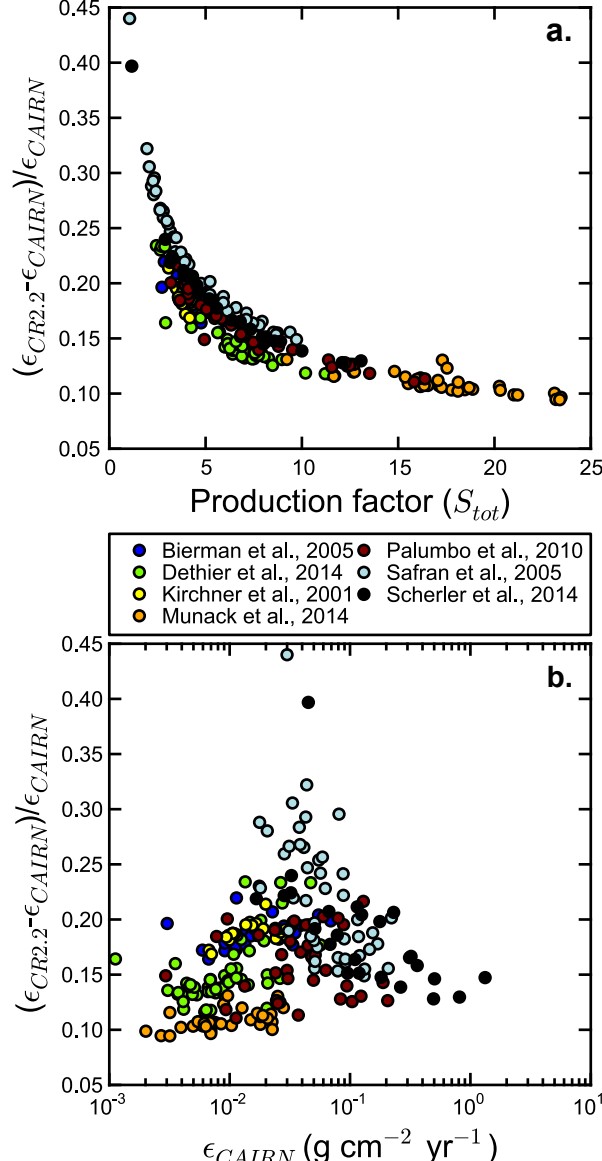

**Figure 6.** Differences between the denudation rate calculated by CAIRN ($\epsilon_{CAIRN}$) and the denudation rate calculated with CRONUS-2.2 ($\epsilon_{CR2.2}$) as a function of CAIRN denudation rate **(a.)**, and differences between the denudation rate calculated by CAIRN ($\epsilon_{CAIRN}$) and the denudation rate calculated with CRONUS-2.2 ($\epsilon_{CR2.2}$) as a function of the total scaling, $S_{tot}$ **(b.)**.

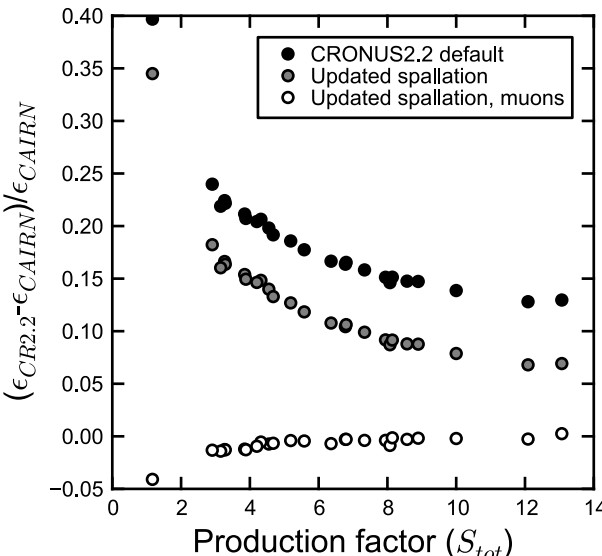

**Figure 7.** Difference between denudation rate calculated by CAIRN ($\epsilon_{CAIRN}$) and the denudation rates calculated by CRONUS-2.2 ($\epsilon_{CR2.2}$), but with CRONUS-2.2. parameters updated to have spallation and muon production reflecting production in CAIRN, which is based on (Braucher et al., 2011). Data are from the Scherler et al. (2014) study.

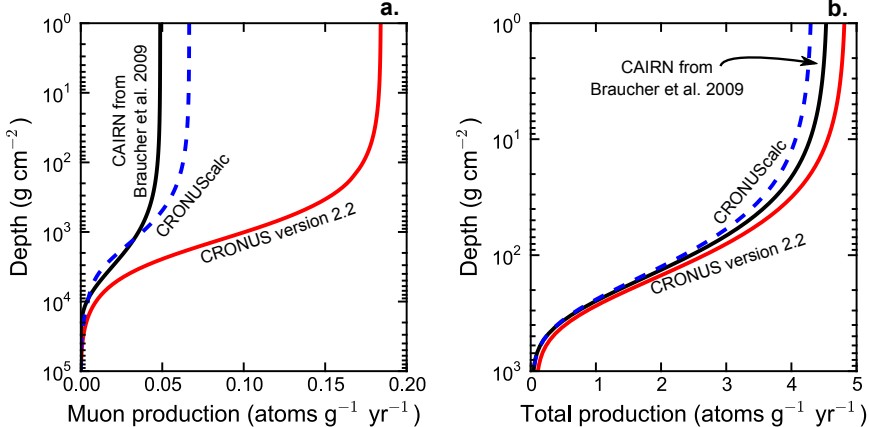

**Figure 8.** Production rates of $^{10}$Be as a function of depth for muons only (**a.**) and total production (**b.**). These production rates are calculated using the Lal/Stone scaling at 70 degrees North and with a pressure of 1007 hPa (near sea level). Note the logarithmic depth scale: eroding particles spend a large amount of their exposure history below 100 g cm$^{-2}$ and so increased muon production at these depths, despite being a small fraction of the total production, plays a significant role in determining the total nuclide concentration (see Figure 9).

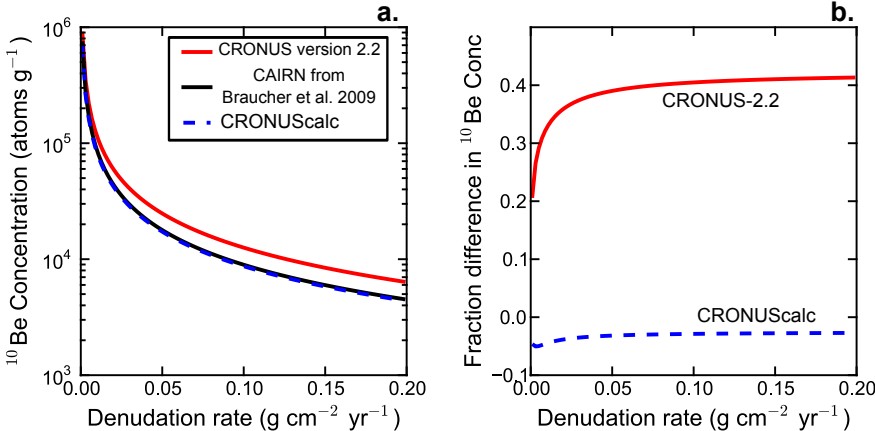

**Figure 9.** Concentrations as a function of denudation rate (**a.**) and the fractional differences between the predicted concentration from the Braucher et al. (2009) approximation used in CAIRN and both CRONUS-2.2 (Balco et al., 2008) and CRONUScalc (Marrero et al., 2016). These concentrations are calculated for a hypothetical site at 70 degrees North and near sea level (1007 hPa). Note that although the default production scheme in CAIRN is the Braucher et al. (2009) scheme, the production from CRONUScalc (Marrero et al., 2016) can also be used (see Table 4).

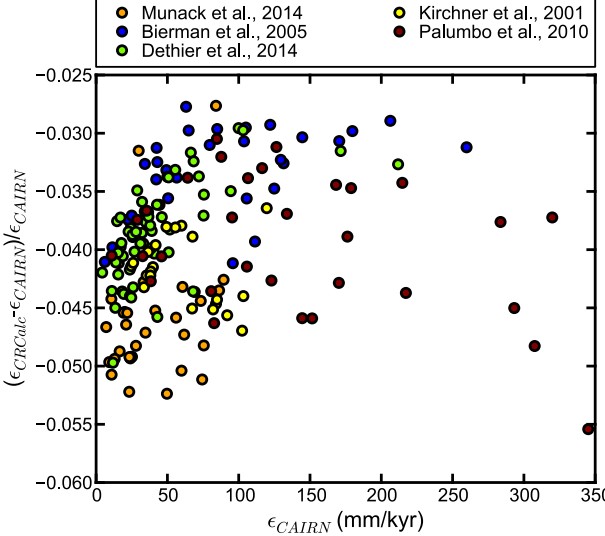

**Figure 10.** Differences between the denudation rate calculated by CAIRN ($\epsilon_{CAIRN}$) and the denudation rate calculated with CRONUScalc ($\epsilon_{CRCalc}$) as a function of CAIRN denudation rate for selected studies.

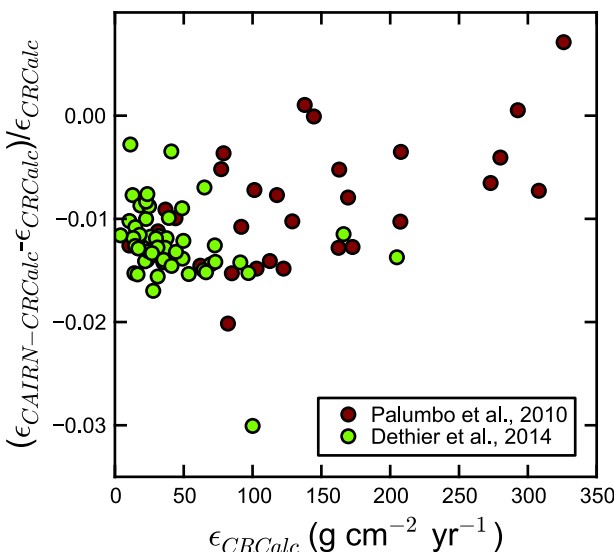

**Figure 11.** Differences between the denudation rate calculated by CAIRN using the parameters in Table 4 to approximate CRONUScalc production ($\epsilon_{CAIRN-CRCalc}$) and the denudation rate calculated with CRONUScalc ($\epsilon_{CRCalc}$) as a function of CAIRN denudation rate for selected studies.

---

**Algorithm 1** Calculating denudation rates on a pixel-by-pixel basis

---

1: Make initial denudation rate guess based on spallation only at outlet pressure and latitude.

2: **repeat**

3:    **for all** Pixels in basin **do**

4:       Calculate cosmogenic nuclide flux based on denudation rate using Eq. 14

5:    **end for**

6:    Average the cosmogenic nuclide concentration over the basin

7:    Change denudation rate by small increment

8:    **for all** Pixels in basin **do**

9:       Calculate cosmogenic nuclide flux based on updated denudation rate using Eq. 14

10:    **end for**

11:    Calculate new denudation rate based on the change in error between calculated and measured cosmogenic nuclide concentrations (i.e., Newton's method).

12:    Calculate change in effective denudation rate

13: **until** Change in effective denudation rate < tolerance

---