# Peer review of "The CAIRN method: Automated, reproducible calculation of catchment-averaged denudation rates from cosmogenic radionuclide concentrations"

_Earth Surface Dynamics, 2016_

## Referee Comment (RC1) · Anonymous Referee #1 · 6 May 2016

The manuscript (ms) by Mudd et al. is a timely and well written contribution aimed at providing the Earth surface community with an open-source tool to calculate consistent and reproducible basin-averaged denudation rates from the cosmogenic nuclide concentrations of stream sediment samples. The ms is well suited for this journal. I anticipate that it will help to foster the determination of reproducible catchment-wide denudation rates in the future, which would be a significant step forward. I have a list of minor comments (see below), which the authors should address.

1. Assumptions of the method: To provide accurate denudation rates from cosmogenic

[Figure]

nuclides, several assumptions must be strictly met. For instance, sediments should be well mixed, there should be a steady state between nuclide production and removal by erosion and decay, a uniform quartz yield etc. These assumptions should be briefly mentioned in the introduction of the ms along with a few references where readers can find more details (e.g. Dunai, 2010; already cited). As in many natural situations the required assumptions are partly violated, the general accuracy of denudation rates is probably limited to $\pm20$ to $\pm30\%$ (Dunai, 2010, p. 124) and this should be mentioned. The general error of 10-20% (mentioned on line 429) is too optimistic in my opinion.

2. Density: To calculate denudation rates in units of length per time one must make an assumption on density (as already pointed out by the Associate Editor). I agree with the editor that the authors should elaborate on this issue in the discussion. From discussions I had in the past, I got the impression that it is not very clear what values are appropriate, because soil thickness varies within catchments and around the globe. Maybe the authors can suggest some recommendations.

3. Nuclide production by muons: The production of muons is mentioned in several places of the ms (e.g. lines 77ff, 298, 507, Fig. 9). As it is known since many years that the model of Heisinger overestimates muon production (shown by the depth-profiles published by Braucher et al. and other studies, which are already cited in the ms), I suggest to make this point clear from the beginning of the ms and not only at the end (507ff). Hence, on line 79 the authors could rephrase the respective sentence to "... field-based estimates of muon production demonstrate that Heisinger et al..." or something similar. Likewise, lines 297-300 should be rephrased to provide a clear picture of that issue.

Balco et al. (2008) provide a nice plot (their Fig. 8), which highlights that the importance of muons relative to neutrons depends on elevation and the rate of erosion. Muons are particularly important at low elevation and at high erosion rates. I think it would be worth stating this more clearly somewhere in the ms.

4. Landsliding: Landsliding introduces considerable complications for interpreting cosmogenic nuclide concentrations in terms of denudation rates (see also comment made by Associate Editor). It seems to me that the approach which the authors propose in order to deal with the issue is too simplistic. In fact, they state on line 324 that their "landsliding module is admittedly rudimentary". Hence, I suggest to omit the respective parts from the ms (i.e. lines 149-158 and 251-254). It is sufficient to mention the landsliding issue and cite a few relevant studies (as the authors have done). I do not think that this will weaken the paper in any way.

5. Snow shielding: It is not yet widely acknowledged that water or snow have a significantly shorter neutron attenuation length than rocks (the value for the latter is ∼160 g/cm2). Therefore, I appreciate that the authors cite the work of Delunel et al. (2014) and Zweck et al (2013). In addition, I suggest to mention the value of 109 g/cm2 for the neutron attenuation length of snow explicitly (cf. Zweck et al. 2013). Fig. 7 of Delunel et al. (2014) shows that the attenuation by snow may be even more significant, which could also be mentioned. In my opinion, it would strengthen the numerical code, if a lower attenuation length of snow (as compared to rock) would be implemented into CAIRN (and the user can thus make a choice). Remote sensing will most likely be increasingly used to map snow depth in mountains (e.g. Beniston et al. 2003 and references therein). Could the authors check the literature and cite 1-2 recent papers on this subject in lines 330ff (I am quite sure there are more recent studies than Beniston et al.).

Beniston et al. (2003). Snow pack in the Swiss Alps under changing climatic conditions: an empirical approach for climate impacts studies Theor. Appl. Climatol. 74, 19–31 (2003) DOI 10.1007/s00704-002-0709-1

6. Hardware/software requirements and standardization: Are there any hardware or software requirements for running the code? (as the Associate Editor downloaded the code and it appeared to work, I did not check it myself). If yes, this should be described somewhere (maybe near line 59). When using the CronusEarth Online calculator, one

has to chose an AMS standard. The authors should mention whether CAIRN has the same option.

7. Change of production rate by seemingly ~20 % (line 453ff): The 20% change in production rate mentioned here gives a somewhat negative impression. In fact, about half of this change is related to a new standardization, which was required after the study of Nishiizumi et al. (2007). In other words, a 10Be exposure age (or erosion rate) calculated with the OLD standardization and a production rate of ~5.0 at/g/yr yields basically the same age (or erosion rate) as a 10Be age calculated with the NEW standardization and a production rate of ~4.5 at/g/yr. Details can, for instance, be found on the website of PRIME Lab (Purdue University) at: http://www.physics.purdue.edu/primelab/News/news0907.php. This issue should be clarified.

8. In section 2.2, it may be useful to provide the simple equation 11 of Lal (1991) in the form: Denudation rate = (Prod. rate / Conc. – lambda) (attenuation length / density)

COMMENTS tied to line numbers Line 63: The term "solution of CRN" in the title is a bit strange. One could rephrase e.g. as "Deriving/quantifying denudation rates at a single location" or something similar. Line 71: maybe insert "local" or "site-specific" before production rate. Line 113: I see that d is shielding depth, but what exactly is d0? (maybe I overlooked it). Line 274: Typo, propAgation. Line 294: Here a production ratio of 6.1 is mentioned. What does the factor of 1.106 mean? The 10Be and 26Al production rates given in Table 3 imply a production ratio of 7.2. Can the authors explain the reasons for this discrepancy? Line 413: Typo; to infer a(N) denudation rate. Line 434: replace "geomagnetic" by "time-dependent". Line 440: please refer to one of the studies by Riebe et al. (2001, 2003). Riebe, C.S. Kirchner, J.W., Finkel, R.C. (2003). Long-term rates of chemical weathering and physical erosion from cosmogenic nuclides and geochemical mass balance. Geochimica et Cosmochimica Acta, 67, 4411-4427. Riebe, C.S. Kirchner, J.W., Granger, D.E. (2001). Quantifying quartz enrichment and its consequences for cosmogenic measurements of erosion rates from

alluvial sediment and regolith. Geomorphology, 40, 15-19.

Line 503: Typo; "denudation" is spelled wrongly. Line 530: I guess "differences" would be more appropriate than the term "errors", which is used twice in this line.

FIGURES a) At the beginning of the captions for Figs. 2, 3, 7, 8, 11, 12 the term "Errors" is used. I believe that the word "Differences" would be more appropriate.

b) The colors of the symbols used in Figs 2, 3, 4 etc for the individual studies are inconsistent (i.e. different colors are used in different plots for one and the same study). Shouldn't the color coding be consistent?

c) I agree with the Associate Editor that some of the figures could be combined.

d) Fig. 5 needs to be increased in size.

TABLES As the first part of the ms is mainly focussed on the description of the equations parameters etc and the data from selected studies are only discussed later, I suggest to reverse the order of Table 2 and Tables 3, 4.

Table 3: Should not the Braucher et al. 2011 EPSL paper be mentioned here (instead of Braucher et al. (2009)? It is the 2011 paper, which gives the SLHL muon prod. rates in Table 6.

Table 3 and 4: I suggest to also provide the absolute muon production rates at SLHL (not only the F values).

Algorithm 1: should the ">" not be reversed to "<" ?
* * *

---

## Referee Comment (RC2) · G. Balco (Referee) · 8 Jun 2016

Review of 'The CAIRN method...' by Mudd and several others.

Summary: This work is very useful and I'm quite supportive of it. The rest of the review below contains (i) some contextual remarks; (ii) a list of areas of the paper where I could not exactly understand what is happening, so clarification is needed; and (iii) a few other comments.

Contextual remarks: Basically, this paper describes a numerical scheme to compute basin-scale erosion rates (henceforth, 'E') from cosmogenic-nuclide concentrations

(henceforth, '$\bar{N}$') in stream sediment. The basic concept here is that given a number of assumptions, these two have a simple inverse relationship that is defined by the mean production rate of the nuclide in question in the basin (henceforth, '$\bar{P}$'), the nuclide's decay constant, and the apparent e-folding length for subsurface production (henceforth, '$\Lambda$'). If we know these three things we can relate E to $\bar{N}$ very simply.

If we accept the assumptions, there are really only two problems with this method overall. The easy one is just calculating $\bar{P}$. Because P is nonlinear with elevation, the point in the basin that has the average elevation doesn't have the average P. So you can define the elevation (and topographic and other shielding, if you want to get fancy) of each pixel in a DEM of the basin, calculate P in each pixel, and take the average. If you want to reverse-engineer some code that has elevation rather than $\bar{P}$ as input, you can also represent the basin by a single point with an effective elevation and/or shielding such that P in that pixel is the same as $\bar{P}$ for the whole basin. So as long as you have a not-too-terrible DEM of the basin, or at least its hypsometry, then both of those methods are sufficient to solve the problem of estimating $\bar{P}$.

The harder problem is that the true depth dependence of production can't be represented by a single exponential with one value of $\Lambda$, which messes up the nice clean explicit relationship between $\bar{N}$ and E. In fact, it makes the formula implicit so you can't solve it analytically.

To deal with this mess in the 2008 online calculators, we used the represent-the-basin-by-a-single-pixel approach and an implicit solver for that pixel. That's efficient, but kind of lame. The correct way to do the problem is to do the implicit-solver scheme for the entire basin. That is, pick a value of E, do a full forward calculation of the nuclide concentration predicted in each pixel at that erosion rate, and take the average to obtain a predicted value of $\bar{N}$. Then repeat, using some kind of an implicit solver, to obtain the value of E that matches the observed $\bar{N}$. Besides just being the correct way to do the simple problem in which the erosion rate is assumed steady and uniform, this method has the enormously large advantage that you can implement other forward

models involving nonsteady or nonuniform erosion and, potentially, use cosmogenic-nuclide measurements to constrain more complicated things than just a simple steady erosion rate. Obviously, this full-implicit scheme is much more computationally painful, because you have to do the implicit solver for all of what could be a large number of pixels. Although this has been done in the existing literature (e.g., Fox, Leith, and others in EPSL, 2015), it's pretty rare. In any case, the aim of this paper is to apply the correct and better method at a large scale, by applying some actual software engineering expertise to incorporate the capability into a GIS package.

So from the perspective of past approaches to this problem, Simon Mudd and his co-authors have basically shown up with a gun at a knife fight. Adding this tool to a topographic analysis package is really useful, and it is potentially enabling of all sorts of more complex applications of cosmogenic-nuclide measurements that haven't been thought about much so far. Although it's probably more accurate to say they have shown up with a light saber at a knife fight, because for many people using this tool will first require an extended apprenticeship with a Linux Jedi master. But the point is that the method described here is not at all lame, it is actually the correct numerical way to do this, and it is potentially enabling of new science. That's all good.

On the other hand, this work creates a new problem. This is not so much a difficulty with the paper itself as a difficulty with the method itself, but what this work focuses on is a high-precision implementation of a model calculation. The problem here is that the model calculation itself is pretty well known to be a poor representation of reality in most cases, because the geological assumptions on which the model is based are highly unlikely to be true. There are a variety of reasons for this (some of which I'll get into later), but the most important is that the model calculation implemented here assumes that the erosion rate in the basin has been constant for long enough so that the nuclide concentrations at all points (and depths) in the basin have reached an equilibrium between production and nuclide loss due to decay and erosion. Although this might be true for near-surface production at relatively high erosion rates, there is

really no possible way it can be true in the subsurface where production is mostly due to muon interactions (see more discussion below). Thus, there is somewhat of a risk of confusing the precision of the model calculation with the accuracy of the actual erosion rate estimate, and it would be incorrect to argue that the method described here, even if it executes the model calculation more precisely, will allow anyone to more accurately estimate real erosion rates. On the plus side, using the method in this paper to take the numerical difficulties in the model calculation off the table ought to allow people to focus more on the geological uncertainties, which is potentially valuable.

So that is the main point of what I am trying to say here. It is great that someone has finally written nice code to do this problem properly, and incorporating the full-forward-model into a topographic-analysis software package is potentially enabling for new science, but it is important to keep in mind that the accuracy of the erosion rate estimates themselves is unlikely to have been improved. Thus, from the perspective of routine calculations of apparent basin-scale erosion rates, it is not clear that an improvement has been made over existing approximate schemes. The main value of this paper is likely to be in future work that explores how to (i) evaluate the geological assumptions in the model, and (ii) learn about unsteady or nonuniform erosion rates from cosmogenic-nuclide measurements.

End of contextual remarks. The rest of this review goes through the paper in order and highlights locations where either (i) I could not understand what was happening, or (ii) improvements could be made. Also some minor comments are included as well. Note that a weakness of this review is that I am not very fluent in C++, so it is possible that some areas of the text that I found hard to understand are clearer in the code than I think they are.

Abstract, first sentence. The first sentence can be removed – it would be more useful for the abstract to begin with the second sentence, which describes what the authors actually did in the paper.

Line 31. The word 'cosmogenic' means originating from cosmic rays. The calculators don't originate from cosmic rays, so 'cosmogenic calculator' makes no sense. Suggest just 'calculator' here.

Line 35-ish. This is a bit oversimplified. In fact, there is not a lot of variation in approaches to computing the erosion rate (really there are only two: include muons, or don't). What there is a lot of variation in is how to compute $\bar{P}$. Also, the phrase "representative parameters for the catchment" is unnecessarily complicated and also unhelpfully nonspecific. Specifically, what you are trying to estimate is just $\bar{P}$. Be more specific here.

Line 40-ish. Again, this is somewhat misleading as written. The mathematical definition of the catchment-averaged erosion rate (e.g., from Bierman and Steig, etc.) specifically has $\bar{P}$ in it, and there is no doubt about what this parameter is. Thus, it is incorrect to say that authors are trying to "choose a production rate that is 'representative' of the catchment." Instead what they are doing is trying to estimate the mean production rate in the catchment, either by computing that in some pixel-based way or by fudging the input to the online calculators to force them to compute the mean production rate internally. In any case, it would be helpful to clarify this a bit.

Line 53. I was confused by this discussion of the landslide scenario, because this scenario violates the assumptions inherent in computing the catchment averaged erosion rate from a Be-10 concentration. Thus, if you are going to compute the erosion rate using the method in Bierman, Steig, etc., you have already assumed that erosion is steady over time (although not spatially uniform), which is equivalent to stating that landslides do not occur. Thus, landsliding doesn't cause a problem with computing $\bar{P}$ (as is implied by its inclusion in this section), it basically invalidates the entire method. To clear this up, I suggest dividing this section into two subsections: (i) issues that make it difficult to compute $\bar{P}$ (e.g., shielding, snow cover, etc.); and (ii) issues that invalidate the entire method by violating its basic assumptions (landsliding, sediment storage, etc.). Alternatively (and probably better), I suggest just pending the entire discussion of landsliding to a separate section at the end of the paper, in which you can discuss more generally the point that you could potentially use this code for all sorts of nonequilibrium scenarios.

Line 60. Don't you want to continue here to mention that in addition to computing production rates, the software also does the implicit solution for erosion rate given a measured nuclide concentration? Because as written this paragraph doesn't indicate that any erosion rate estimate happens. Needs improvement. In general, also, it would be helpful for the reader at this stage if you were to explain the overall procedure of inverting a forward model for nuclide concentrations to obtain an estimate of the erosion rate, as a preview of coming attractions.

Line 80-ish. This brings up the subject of muons. In this work as in others, muons are responsible for 2% of surface production and 98% of suffering. The decision in this paper to use an exponential scheme for muon production for computational simplicity is, in fact, sensible. Unfortunately I found the explanation here to be incomplete and confusing.

Basically, the difference between the Heisinger integration scheme and a simple exponential approximation à la Braucher is that the latter is incorrectly representing the physics. What is really happening physically is that as depth increases, the mean energy of the remaining muons increases, so the instantaneous e-folding length for muon production continually increases with depth. You can't represent that with a finite sum of exponentials. If you have a bunch of summed exponentials, you can do pretty well at shallow depths, but there is some depth below which you are quite wrong. So that is what the actual difference is.

However, there are two reasons the exponential approximation is OK here.

The first is just that only a very small fraction of the nuclide concentration you are measuring was produced by muons at greater than a few meters depth, so accurately estimating production at that depth doesn't matter much. And when the erosion rate is

high enough to care about this approximation, you have other problems to do with the model assumptions.

The second is that if you include muons in the production rate calculation, you are also including muon production in the basic assumption of the method that the surface has been eroding at a steady rate long enough for the Be-10 concentration to reach production-erosion equilibrium. For muon production, this takes a really long time. It is easiest to think of this as a half-life for approach to equilibrium, which is $-\ln(0.5)/(\lambda + E/\Lambda)$ where $\lambda$ is the decay constant of the nuclide in question (4.99e-7 for Be-10); E is the erosion rate (g/cm$^{2]}$/yr), and $\Lambda$ is the e-folding length for the depth dependence of production (g/cm$^{2]}$). Take a landscape that is eroding at 50 m/Myr. For spallogenic production ($\Lambda \simeq 150$), this half-life is 13,000 years. It is already a bit of a stretch to assume that a typical surface has had an unchanging erosion rate for several times 13,000 years, i.e. from well before the LGM until now. However, for muon production (which has $\Lambda > 1500$) this half-life is at least one order of magnitude longer (> 120,000 years, increasing with depth). It is highly unlikely that any landscape has experienced steady erosion for many times 130,000 years. Because of these different averaging times, tt is also highly likely that the erosion rate recorded by muon-produced nuclides is very different from the erosion rate recorded by the spallogenic inventory (see Dethier and others, Geology, 2014 for a nice example). So including an accurate representation of muons in the calculation would, in principle, improve the accuracy of the erosion rate calculation if the steady state assumption is true. However, it also decreases the likelihood that the steady state assumption is true. For this reason, there is no point getting too worried about how exact the muon production estimate is in this calculation.

To summarize, it is a sensible move to limit the complexity of the muon production calculations because any increase in precision from making them more elaborate is probably illusory. However, this issue should be explained more clearly in the text.

A final point here is that there would potentially be lots of other ways to speed up the computation whilst still using the Heisinger scheme, if you wanted. Mainly this is
because the production rate due to muons will not be very different between adjacent cells. Thus, it is a big waste of time computing muon production separately in each pixel. You can probably get away with doing the muon calculations in a very small minority of cells in a typical watershed and extending those results to the other cells simply by a regression formula in elevation, without loss of accuracy. Or do muon production on a much coarser grid than spallogenic production. Of course, this would be a big rewrite of the code, but if you really want it to run maximum fast it would be the next obvious strategy. If muons are 2% of surface production, why give them more than 2% of processor time?

Line 105. Again, equations are not cosmogenic.

Line 120. Note again the implications of assuming that muon production is in steady state. Not likely to be true.

Line 130, "self-shielding." I am not sure I understand what is going on here because under normal circumstances, the sediment leaving the catchment would be assumed to be from an infinitesimally small surface layer, so no integration in depth would be required. So this appears to me to be overly complex. My understanding of what is happening in this part of the paper is that the authors have just put in this capability to facilitate later use of the same code for a patchy-erosion model where finite thicknesses of sediment are removed at once (e.g., landslides). And then the discussion of steady state is confusing here as well, because, of course, if landslides are occurring then there is by definition not a steady state. Overall, more explanation needed here. As noted above, I think this would be clearer if all discussion related to the landsliding issue was deferred to a separate section.

Line 160 and below. Topographic shielding. This is another example where the calculation is an precise representation of simplified physics, so gives illusory precision. Specifically, this code includes a quite precise calculation of topographic shielding under the assumptions that (i) the cosmic-ray flux is totally attenuated below the apparent

horizon, and (ii) the zenith angle dependence of the cosmic-ray flux responsible for production is a cosine to a constant power. Neither (i) or (ii) is actually true. Because secondary particle production takes place throughout the atmosphere (including that part of the atmosphere that is between you and the apparent horizon on the other side of the valley), a nonzero amount of production will actually be due to cosmic rays originating below the visible horizon. In addition, the cosine-to-a-power dependence is highly approximate. See this paper:

Argento, D.C., Stone, J.O., Reedy, R.C. and O'Brien, K., 2015. Physics-based modeling of cosmogenic nuclides part II–Key aspects of in-situ cosmogenic nuclide production. Quaternary Geochronology, 26, pp.44-55.

The point being that the very comprehensive analysis of discretization errors in the shielding calculation here clouds the fact that there exist larger systematic errors due to simplified physics. This issue has basically no practical relevance to the erosion rate calculation overall (because it is still much less important than violations of the basic method assumptions). However, the authors should note here that they are concerned with the precision of a representation of simplified physics, which may or may not be the same as the precision of the calculation relative to real life.

In addition, this could also be sped up a lot if you really wanted to. The shielding factor doesn't change much between adjacent pixels. In addition, most of the pixels in a landscape don't actually shield anything; nearly all the shielding is due to ridgelines and smaller things that are near you. There is some code that has been kicking around since 2001 here:

http://depts.washington.edu/cosmolab/oldweb/P$_b$y$_G$$IS.html$

that uses this simplification (note that I think the editor's comment on this subject gives a bad link). But although of historical interest, that is beside the point here.

A final important issue here is that it was not clear to me whether telescoping of the

mean free path length on dipping surfaces (see Dunne, also Fig. 5 in Balco, 2014 in Quaternary Geochronology) is included in this calculation.

Line 200 et al. The issue of non-time-dependent vs. time-dependent scaling is actually more important than described here. The reason for that is that production rates are calibrated using data mostly from the last 20,000 years, and the Earth's magnetic field has been stronger than its long-term average during that time. Thus, at erosion rates low enough that the residence time of material in the soil profile is much longer than this (e.g., most normal erosion rates), the use of a non-time-dependent scheme likely creates a systematic error due to an underestimate of the long-term production rate. Basically this is yet another violation of the steady-state assumption. Again, this is a non-issue compared to much bigger issues in the application of this method, but the text is somewhat inaccurate here as written.

Line 215, section 2.6. Unfortunately, I simply don't understand why the calculation described in this section is necessary. I didn't look back at the Vermeesch paper, but if I am remembering correctly this whole procedure was just needed to make the equation relating erosion rate to concentration explicit so it could be solved analytically?? Here you don't need to worry about that, because you are only doing the forward calculation, so why are you doing this? I may not be remembering this correctly, but in other words, it seems to me that all the S's and F's needed for Equation 13 are known a priori, or should be. Typically one would compute scaling for spallogenic production and muons separately (for example, this is in the Stone (2000) scheme as published), and because they are different, that should take care of the fact that muons are less important at higher elevation. Each pathway has already been assigned its own attenuation length, so you can compute mass shielding for each pathway. Then it seems like all you need to do is decide how to compute topographic shielding for muons (I don't know the answer...in the 2008 online calculators it is just disregarded), and you are done. What am I missing here?

In any case, this needs to be better explained.

Line 270-ish. I should point out (in response to the editor's comments) that the issue of asymmetry of uncertainty distributions is really a total non-issue from the geological perspective. Anything with an e in it will have an asymmetric uncertainty distribution, of course, but it's hard to think of any cases where it's actually important from the geological perspective.

Line 280-ish. Numerical partial differentiation by repeatedly doing the full calculation is almost certainly overkill (especially because you've already linearized it). I would do this simply by assuming that the basin has one pixel with the effective $\bar{P}$ derived from the whole basin. I agree that it is interesting to do it once, though.

Line 300. In physical science, 'conservative' is typically used to mean that something is being conserved, e.g., mass or energy. This use in the context of uncertainty analysis is common but incorrect. Instead one should state that the uncertainty estimate is supposed to be an upper bound.

Line 315. This is an excellent point, that the divergence among various theoretical expectations of how snow shielding works is much less important than the practical difficulty of actually measuring the mean snow depth distribution throughout the year. Frankly, in my view it is not even necessary to mention the various models here, because that issue is pretty much totally unrelated to this paper. As an aside, I found the Delunel paper to not be persuasive because, as far as I can tell from reading the paper, we don't know the effect of snow cover on the energy dependence of the neutron monitors (snow is basically like changing the amount of polyethylene on the outside, which affects the spectral response). This would imply that it is likewise unknown whether or not the variation in monitor count rate with snow cover is applicable to cosmogenic-nuclide production at all. But that is totally off topic.

Line 320-ish. Again, I suggest moving all discussion related to landsliding and non-steady/nonuniform erosion to a separate section at the end. First, implement the basic model; then, at the end, introduce the abilities to deal with complications that are halfbaked at present, but potentially useful in future.

Line 345-ish. It is probably overkill to generate separate effective elevations and average shielding factors for input to the 2008 online calculator. I know that technically it's required because the elevation affects the muon proportion of total production and the shielding factor doesn't, but this issue is well down in the noise.

Line 370. Again, no need to get into the details of snow shielding here. The point remains that it is unclear whether one can estimate the snow depth accurately in any case.

In general, in this part of the discussion (i.e., all of section 5) I think it would be easier to understand if you break down the discussion into two parts: things that are linear with respect to the production rate (e.g., topographic shielding), so can be pixel-averaged by themselves; and things that are nonlinear with respect to the production rate (e.g., elevation, latitude, snow shielding), that have to be converted into production rates, averaged, and then unconverted into an effective summary value. At present these two things are mixed up and it's hard to understand what is happening. Overall, this section could be made more clear.

Section 6. This sort of comparison is a terrible mess because of the need to sort out the differences between inherent properties of the algorithm (e.g., point approximation vs. full-basin calculation) and the input parameters (mainly the production rate and the muon interaction parameters). Overall, however, it is accomplished fairly well in this paper; I like the approach of selecting a few representative data sets rather than trying to show a global comparison over all of scaling and erosion rate space, and the explanation is quite clear as regards which errors apply where in which comparisons. I only have a couple of comments about this section.

Line 470. It is quite interesting that there is a systematic difference in shielding factors vs. those originally reported. Do you think this really is because of the averaging-a-nonlinear-thing effect? But in any case, as discussed above, the precision of this

measurement is overstated in any case due to simplification of the physics.

Line 480. I think production rate differences are much more likely responsible here than anything to do with shielding calculations.

Line 486. The snapping issue is by far the biggest problem I can think of in wholesale automation of this process. Especially potentially disastrous for literature data.

Line 500+ and Figures 7-8. As noted by the other reviewer, this effect is nearly all due to differences in the input parameters (production rate and muon interaction cross-sections) and very little due to the spatial-averaging issue or any other aspects of the various algorithms. In large part this is my fault because I have been too much of a slacker to update these parameters in the online code (that is, make v 2.3), which is, frankly, embarrassing. Sorry. However, the need here for the purposes of this paper is to clearly separate these issues. I can think of two ways to do this. One, change the parameters in the CAIRN code to increase spallogenic production by a factor of (4.5/4) and muon production by a factor of (1/0.44). That will very nearly account for the various input parameter differences. Then do the comparison on that basis. Two, leave these figures unchanged but add in the background some lines showing the expected effect of those changes in the parameters (for elevations vaguely resembling the input data). In any case, this would be extremely helpful in distinguishing the various effects of differences in the algorithm itself vs. differences in the calibrated parameters. This would also make the discussion in lines 520+ more clear as well.

Besides the above comments on the MS, I have a couple of comments on another review ("interactive comment" by anonymous reviewer 1).

1. Reviewer's comment 3. It is not correct to say that 'the model of Heisinger overesti-mates muon production.' The model accurately estimates muon fluxes; the problem is that the cross-sections for production of Be-10 and Al-26 by muon interactions, when applied to those flux estimates, overestimate Be-10 and Al-26 production. In other words, it's not the model that's wrong, it's the cross-section measurements needed to

convert the model prediction to a production rate. I think the present paper is mostly correct on this point.

2. Reviewer's comment 7. The reviewer is correct here, and this is important. The options here are (i) to get all numbers properly standardized in this discussion, or (ii) preferably, to not get into the details here and simply note that best estimates of production rates are about 10% lower than they were 10 years ago because of improved calibration data.

Finally, some final remarks:

Use of acronym 'CRN.' Do you really want to exclude stable cosmogenic nuclides? Because this code would work for them too. Perhaps, having written the paper, the authors could just globally search and replace 'CRN' with 'cosmogenic nuclide?'

'CAIRN' acronym. This is really beside the point with respect to the scientific content of the paper, but when you have to remove important words ('cosmogenic') and randomly select letters from the middle of other words to eventually wind up with a word that appears in the dictionary, I question whether things have really been improved. If we are allowed to select randomly from all the letters in 'Catchment-averaged denudation rates from cosmogenic nuclides' as long as we maintain their original order, why not 'CRIMES,' 'EVIL,' 'CAGE-FREE,' or 'ACNE' (I favor 'CAGE-FREE')? Certainly the authors are entitled to call their code whatever they want, but why such a desperate bid for an acronym at any cost? Especially as it's already a part of something with a different name (LSDTopoTools). Perhaps we could just call it the cosmogenic-nuclide module of LSDTopoTools?

---

## Author Comment (AC1) · 7 Jul 2016

We thank reviewer 1 for their comments, which have helped us improve the paper. Our responses are in *blue italics*.

**Reviewer 1**

1. Assumptions of the method: To provide accurate denudation rates from cosmogenic nuclides, several assumptions must be strictly met. For instance, sediments should be well mixed, there should be a steady state between nuclide production and removal by

erosion and decay, a uniform quartz yield etc. These assumptions should be briefly mentioned in the introduction of the ms along with a few references where readers can find more details (e.g. Dunai, 2010; already cited). As in many natural situations the required assumptions are partly violated, the general accuracy of denudation rates is probably limited to $\pm 20$ to $\pm 30$ % (Dunai, 2010, p. 124) and this should be mentioned. The general error of 10-20% (mentioned on line 429) is too optimistic in my opinion.

*The second reviewer also raises this point. We agree that the computation invokes a number of assumptions (mentioned above) that are almost certainly not met and we do not wish to deceive users of the code that the results are more accurate than they in fact are. On the other hand, it is difficult to quantify the uncertainties generated by these assumptions. We thus have added a new section to make it clear that the general error is a minimum error that captures only known errors in AMS uncertainty, production rate uncertainty, etc. This new section follows the advice of reviewer 2.*

2. Density: To calculate denudation rates in units of length per time one must make an assumption on density (as already pointed out by the Associate Editor). I agree with the editor that the authors should elaborate on this issue in the discussion. From discussions I had in the past, I got the impression that it is not very clear what values are appropriate, because soil thickness varies within catchments and around the globe. Maybe the authors can suggest some recommendations.

*We certainly agree with this comment and there is no good way to arrive at a density estimate of soils if you do not have measurements. However, in the discussion we have used density values for rock, which is less spatially variable than soil density, and thus the erosion rates reported in length per time are rock equivalent erosion rates (surface erosion rates will be significantly higher since soil is less dense). We now state this explicitly in the manuscript. In the summary of calculations section we now state explicitly that CAIRN reports denudation rates in g $cm^{-2}$ $yr^{-1}$ because this requires no assumption of density, and then in the discussion we add text with notes on what density means. We have seperated this section on density in the discussion. Some figures still report denudation rates in length per time in order to remain consistent with*

*rates reported in other papers.*

3. Nuclide production by muons: The production of muons is mentioned in several places of the ms (e.g. lines 77ff, 298, 507, Fig. 9). As it is known since many years that the model of Heisinger overestimates muon production (shown by the depth-profiles published by Braucher et al. and other studies, which are already cited in the ms), I suggest to make this point clear from the beginning of the ms and not only at the end (507ff). Hence, on line 79 the authors could rephrase the respective sentence to "... field-based estimates of muon production demonstrate that Heisinger et al..." or something similar. Likewise, lines 297-300 should be rephrased to provide a clear picture of that issue.

*We have stated more clearly on the former lines 77ff and 297ff that field studies show the Heisinger et al. model overestimates muon production, but have not made major changes based on the comments of reviewer 2.*

Balco et al. (2008) provide a nice plot (their Fig. 8), which highlights that the importance of muons relative to neutrons depends on elevation and the rate of erosion. Muons are particularly important at low elevation and at high erosion rates. I think it would be worth stating this more clearly somewhere in the ms.

*We now state this.*

4. Landsliding: Landsliding introduces considerable complications for interpreting cosmogenic nuclide concentrations in terms of denudation rates (see also comment made by Associate Editor). It seems to me that the approach which the authors propose in order to deal with the issue is too simplistic. In fact, they state on line 324 that their "landsliding module is admittedly rudimentary". Hence, I suggest to omit the respective parts from the ms (i.e. lines 149-158 and 251-254). It is sufficient to mention the landsliding issue and cite a few relevant studies (as the authors have done). I do not think that this will weaken the paper in any way.

*We have followed the advice of reviewer 2 and now have a section devoted to transient scenarios. We removed text about landsliding from the former line 251 and 324.*

5. Snow shielding: It is not yet widely acknowledged that water or snow have a significantly shorter neutron attenuation length than rocks (the value for the latter is $\pm 160$ g/cm$^2$). Therefore, I appreciate that the authors cite the work of Delunel et al. (2014) and Zweck et al (2013). In addition, I suggest to mention the value of 109 g/cm$^2$ for the neutron attenuation length of snow explicitly (cf. Zweck et al. 2013). Fig. 7 of Delunel et al. (2014) shows that the attenuation by snow may be even more significant, which could also be mentioned. In my opinion, it would strengthen the numerical code, if a lower attenuation length of snow (as compared to rock) would be implemented into CAIRN (and the user can thus make a choice). Remote sensing will most likely be increasingly used to map snow depth in mountains (e.g. Beniston et al. 2003 and references therein). Could the authors check the literature and cite 1-2 recent papers on this subject in lines 330ff (I am quite sure there are more recent studies than Beniston et al.).

*We have now cited the Beniston paper and mention that Zweck has a reduced attenuation thickness in the section "Spatial averaging for the CRONUS calculators". There are papers more recent than Beniston but these mainly use GCMs to model changing snow conditions and we feel scientists who want to calculate denudation rates are unlikely to use such models to reconstruct snow thicknesses. We also refer to the comment of reviewer 2: the uncertainties in the snow thickness through time vastly outweighs the uncertainties in attenuation lengths so we do not feel that changing the code in regard to snow calculations will improve accuracy of the model. We do however add text stating that users can replicate the changing attenuation thickness suggested by Zweck by modifying the snow raster fed to CAIRN.*

6. Hardware/software requirements and standardization: Are there any hardware or software requirements for running the code? (as the Associate Editor downloaded the code and it appeared to work, I did not check it myself). If yes, this should be described somewhere (maybe near line 59). When using the CronusEarth Online calculator, one has to chose an AMS standard. The authors should mention whether CAIRN has the same option.

*Indeed CAIRN forces users to choose the AMS standard, we have now stated this in the section on "Summary of CAIRN parameters for denudation calculations". We have also included a short note on software/hardware requirements on the former line 59.*

7. Change of production rate by seemingly $\pm 20\%$ (line 453ff): The 20% change in production rate mentioned here gives a somewhat negative impression. In fact, about half of this change is related to a new standardization, which was required after the study of Nishiizumi et al. (2007). In other words, a 10Be exposure age (or erosion rate) calculated with the OLD standardization and a production rate of $\pm 5.0$ at/g/yr yields basically the same age (or erosion rate) as a 10Be age calculated with the NEW standardization and a production rate of $\pm 4.5$ at/g/yr. Details can, for instance, be found on the website of PRIME Lab (Purdue University) at: http://www.physics.purdue.edu/primelab/News/news0907.php. This issue should be clarified.

*Good point and also raised by reviewer 2. We have followed that reviewer's advice and simply reported that production rates have changed along with AMS standards over the last decade, and we have also included a figure that shows the differences between CAIRN and CRONUS2.2 are mainly due to different parameter values rather than factors related to the underlying mathematics.*

8. In section 2.2, it may be useful to provide the simple equation 11 of Lal (1991) in the form: Denudation rate = (Prod. rate / Conc. – lambda) (attenuation length / density)

*We have added a modified version of this equation at the end of this section and have explained why it is inadequate for a catchment averaged calculation.*

Line 63: The term "solution of CRN" in the title is a bit strange. One could rephrase e.g. as "Deriving/quantifying denudation rates at a single location" or something similar.

*We have renamed this section "Quantifying denudation rates at a single location."*

Line 71: maybe insert "local" or "site-specific" before production rate.

*Added the word "local".*
Line 113: I see that d is shielding depth, but what exactly is d0? (maybe I overlooked it).
*It is the initial shielding depth, now stated in the manuscript.*

Line 274: Typo, propAgation.
*Fixed.*

Line 294: Here a production ratio of 6.1 is mentioned. What does the factor of 1.106 mean? The 10Be and 26Al production rates given in Table 3 imply a production ratio of 7.2. Can the authors explain the reasons for this discrepancy?
*Thanks for highlighting this: we did not do a very good job of explaining what we were doing in the previous version of the manuscript. We have corrected this passage to reflect what is actually happening in the code (basically we have a fixed production uncertainty of 8.7% which is taken from CRONUS2.2, but which we apply to our updated production rates.*

Line 413: Typo; to infer a(N) denudation rate.
*Fixed.*

Line 434: replace "geomagnetic" by "time-dependent".
*Done.*

Line 440: please refer to one of the studies by Riebe et al. (2001, 2003). Riebe, C.S. Kirchner, J.W., Finkel, R.C. (2003). Long-term rates of chemical weathering and physical erosion from cosmogenic nuclides and geochemical mass balance. Geochimica et Cosmochimica Acta, 67, 4411-4427. Riebe, C.S. Kirchner, J.W., Granger, D.E. (2001). Quantifying quartz enrichment and its consequences for cosmogenic measurements of erosion rates from alluvial sediment and regolith. Geomorphology, 40, 15-19.
*We have included the 2001 Riebe et al paper.*

Line 503: Typo; "denudation" is spelled wrongly.
*Fixed.*

Line 530: I guess "differences" would be more appropriate than the term "errors", which is used twice in this line.
*Changed the wording from error to difference here and elsewhere.*

FIGURES a) At the beginning of the captions for Figs. 2, 3, 7, 8, 11, 12 the term "Errors" is used. I believe that the word "Differences" would be more appropriate.
*We agree this is a better word and have made these changes.*

b) The colors of the symbols used in Figs 2, 3, 4 etc for the individual studies are inconsistent (i.e. different colors are used in different plots for one and the same study). Shouldn't the color coding be consistent?
*Done.*

c) I agree with the Associate Editor that some of the figures could be combined.
*We have combined the former Figures 2 and 3, and the former Figures 7 and 8.*

d) Fig. 5 needs to be increased in size.
*Done.*

TABLES As the first part of the ms is mainly focussed on the description of the equations parameters etc and the data from selected studies are only discussed later, I suggest to reverse the order of Table 2 and Tables 3, 4.
*We have changed the order so the former Table 2 (the default parameters) appears before the data from selected study sites. Table 4 remains in the same place since we felt it only makes sens in the context of comparison of the different calculators, which sets the ordering of the tables.*

Table 3: Should not the Braucher et al. 2011 EPSL paper be mentioned here (instead of Braucher et al. (2009)? It is the 2011 paper, which gives the SLHL muon prod. rates in Table 6.
*Fixed.*

Table 3 and 4: I suggest to also provide the absolute muon production rates at SLHL

(not only the F values).
*We have not made this change since it is simply one line multiplied by another and therefore redundant.*

Algorithm 1: should the $>$ not be reversed to $<$ ?
*We are glad you caught that! Fixed.*
* * *

---

## Author Comment (AC2) · 12 Jul 2016

We thank Dr Balco for his detailed review, which has helped us significantly improve the paper. Dr Balco begins his review with a number of contextual remarks that have helped us refine the paper. We won't reiterate all of these remarks here but instead will try to include responses to the components most obviously requiring modifications of the manuscript. Dr Balco's review was clearly carefully crafted, adds to the discourse surrounding computation of denudation rates based on cosmogenic nuclides, and provides evidence for why an open review process can be beneficial to the scientific process.

Pages C1 and C2 of the review provide contextual comments which we do not believe require a response, but we have adopted some of the language here in our correction of the former line 35 because it is a better statement of the problem than we had in the original manuscript.

Page C3 makes allusions to western cinema and the Star Wars franchise, which we very much appreciate. We are also glad the reviewer thinks our contribution is, in his words, not lame. The bottom of this page and page C4 goes on to discuss the dubious nature of assumptions required to calculate denudation rates, mirroring quite closely the comments of reviewer 1, in their first item. Please see our response to this reviewer about the various assumptions about the natural system we try to model and why they are imperfect. Specifically, we now have a section on how temporal and spatial variations in denudation rates add significantly to the uncertainty of the method, noting that this is not specific to our method, but rather endemic to all estimates of catchment averaged erosion.

Below are responses to itemized queries by the reviewer. Our responses are in *blue text*.

Abstract, first sentence. The first sentence can be removed – it would be more useful for the abstract to begin with the second sentence, which describes what the authors actually did in the paper.
*Sentence deleted.*

Line 31. The word 'cosmogenic' means originating from cosmic rays. The calculators don't originate from cosmic rays, so 'cosmogenic calculator' makes no sense. Suggest just 'calculator' here.
*Done.*

Line 35-ish. This is a bit oversimplified. In fact, there is not a lot of variation in approaches to computing the erosion rate (really there are only two: include muons, or don't). What there is a lot of variation in is how to compute P. Also, the phrase "representative parameters for the catchment" is unnecessarily complicated and also unhelpfully nonspecific. Specifically, what you are trying to estimate is just P. Be more specific here.

*We have changed the text here to mirror what the reviewer has written on page C2 of his review, which succinctly states the process involved in calculating denudation rates.*

Line 40-ish. Again, this is somewhat misleading as written. The mathematical definition of the catchment-averaged erosion rate (e.g., from Bierman and Steig, etc.) specifically has P in it, and there is no doubt about what this parameter is. Thus, it is incorrect to say that authors are trying to "choose a production rate that is 'representative' of the catchment." Instead what they are doing is trying to estimate the mean production rate in the catchment, either by computing that in some pixel-based way or by fudging the input to the online calculators to force them to compute the mean production rate internally. In any case, it would be helpful to clarify this a bit.

*We have rewritten this section, it now says: "Production rates vary spatially, thus users of online calculators must calculate the effective production rate within a catchment using a weighted mean of the production in individual pixels. The manner in which these are fed to existing calculators vary, for example one must feed a single weighted mean production, after shielding corrections to COSMOCALC. In contrast, one must calculate weighted mean shielding corrections and pass them to CRONUS-2.2, and in addition must calculate a pressure or elevation that reproduces the mean production rate before shielding."*

Line 53. I was confused by this discussion of the landslide scenario, because this scenario violates the assumptions inherent in computing the catchment averaged erosion rate from a Be-10 concentration. Thus, if you are going to compute the erosion rate using the method in Bierman, Steig, etc., you have already assumed that erosion is steady over time (although not spatially uniform), which is equivalent to stating that

landslides do not occur. Thus, landsliding doesn't cause a problem with computing P (as is implied by its inclusion in this section), it basically invalidates the entire method. To clear this up, I suggest dividing this section into two subsections: (i) issues that make it difficult to compute P (e.g., shielding, snow cover, etc.); and (ii) issues that invalidate the entire method by violating its basic assumptions (landsliding, sediment storage, etc.). Alternatively (and probably better), I suggest just pending the entire discussion of landsliding to a separate section at the end of the paper, in which you can discuss more generally the point that you could potentially use this code for all sorts of nonequilibrium scenarios.

*We have chosen the latter option suggested here. We now append a separate section on transience and remove mention of landsliding from other parts of the manuscript.*

Line 60. Don't you want to continue here to mention that in addition to computing production rates, the software also does the implicit solution for erosion rate given a measured nuclide concentration? Because as written this paragraph doesn't indicate that any erosion rate estimate happens. Needs improvement. In general, also, it would be helpful for the reader at this stage if you were to explain the overall procedure of inverting a forward model for nuclide concentrations to obtain an estimate of the erosion rate, as a preview of coming attractions.

*Yes, now that you mention it, we do want to say that. The text now reads "Based on these calculations the software can then calculate the expected cosmogenic nuclide concentration from a basin given a spatially homogenous denudation rate. Finally, the software uses Newton iteration to calculate the denudation rate that best reproduces the measured cosmogenic nuclide concentration."*

Line 80-ish. This brings up the subject of muons. In this work as in others, muons are responsible for 2% of surface production and 98% of suffering. The decision in this paper to use an exponential scheme for muon production for computational simplicity is, in fact, sensible. Unfortunately I found the explanation here to be incomplete and confusing.

[Figure]

*We have attempted to follow the reviewer's comments within our manuscript, since they contain succinct statements of why the muon approximation may erroneous and also why errors in muogenic production do not play a significant role in overall uncertainty. See specific changes below.*

Basically, the difference between the Heisinger integration scheme and a simple exponential approximation à la Braucher is that the latter is incorrectly representing the physics. What is really happening physically is that as depth increases, the mean energy of the remaining muons increases, so the instantaneous e-folding length for muon production continually increases with depth. You can't represent that with a finite sum of exponentials. If you have a bunch of summed exponentials, you can do pretty well at shallow depths, but there is some depth below which you are quite wrong. So that is what the actual difference is.

*We now specifically state this: "The advantage of the Heisinger et al. (2002) scheme is that it tries to capture the physics of muon passage through the near surface, and specifically models how the mean energy of muons increases as one moves to greater depths in the subsurface. This affects muon production at depth in a way that is not captured by exponential approximations. Recent work by Marrero et al. (2016) has updated the scheme of (Heisinger et al., 2002) reflecting the muon production rates inferred from field studies. This method still has the disadvantage that it is computationally expensive, to the extent that this computational cost is prohibitive if one is to calculate muon production in numerous pixels across a catchment."*

However, there are two reasons the exponential approximation is OK here. (The reviewer goes on to state why the approximation is okay).

*We have added some text reflecting the reviewer's comments about why the approximation is okay, which hopefully will encourage skeptical readers to keep going: "Our approach is to approximate muon production using a sum of exponential functions. This approach has the advantage of being computationally efficient, but has the disadvantage of not reflecting the physics of muon production and therefore failing to capture*

*muon production well at depths beyond a few meters. This is unlikely to lead to large errors, however, because muon production makes up a very small percentage of the overall nuclide production at the depths where the physics-based models diverge from the exponential models. We specifically quantify this difference in Section 6.3, finding the exponential approximation to lead to differences between the physics-based approximation that are relatively small (for a wide range of denudation rates these differences are less than 2%)."*

A final point here is that there would potentially be lots of other ways to speed up the computation whilst still using the Heisinger scheme, if you wanted. Mainly this is because the production rate due to muons will not be very different between adjacent cells. Thus, it is a big waste of time computing muon production separately in each pixel. You can probably get away with doing the muon calculations in a very small minority of cells in a typical watershed and extending those results to the other cells simply by a regression formula in elevation, without loss of accuracy. Or do muon production on a much coarser grid than spallogenic production. Of course, this would be a big rewrite of the code, but if you really want it to run maximum fast it would be the next obvious strategy. If muons are 2% of surface production, why give them more than 2% of processor time?

*These are all interesting suggestions for speeding up the Heisinger approximations, but our testing suggests the difference between the Heisinger method and our exponential method is around 2%, which is completely dwarfed by the rest of the uncertainties, so we don't think it would be particularly useful to spend the time optimizing this part of the code.*

Line 105. Again, equations are not cosmogenic.
*Removed "cosmogenic".*

Line 120. Note again the implications of assuming that muon production is in steady state. Not likely to be true.
*We now refer readers to our section on transience here.*

Line 130, "self-shielding." I am not sure I understand what is going on here because under normal circumstances, the sediment leaving the catchment would be assumed to be from an infinitesimally small surface layer, so no integration in depth would be required. So this appears to me to be overly complex. My understanding of what is happening in this part of the paper is that the authors have just put in this capability to facilitate later use of the same code for a patchy-erosion model where finite thicknesses of sediment are removed at once (e.g., landslides). And then the discussion of steady state is confusing here as well, because, of course, if landslides are occurring then there is by definition not a steady state. Overall, more explanation needed here. As noted above, I think this would be clearer if all discussion related to the landsliding issue was deferred to a separate section.

*We now state explicitly that for most applications an infinitesimal layer will be used ($d_t = 0$, but we have included it so that future users can devise clever ways to explore landsliding. We then state that landsliding is beyond the scope of this paper, but we acknowledge the uncertainties it introduces.*

Line 160 and below. Topographic shielding. This is another example where the calculation is an precise representation of simplified physics, so gives illusory precision. Specifically, this code includes a quite precise calculation of topographic shielding under the assumptions that (i) the cosmic-ray flux is totally attenuated below the apparent horizon, and (ii) the zenith angle dependence of the cosmic-ray flux responsible for production is a cosine to a constant power. Neither (i) or (ii) is actually true. Because secondary particle production takes place throughout the atmosphere (including that part of the atmosphere that is between you and the apparent horizon on the other side of the valley), a nonzero amount of production will actually be due to cosmic rays originating below the visible horizon. In addition, the cosine-to-a-power dependence is highly approximate. See this paper: Argento, D.C., Stone, J.O., Reedy, R.C. and O'Brien, K., 2015. Physics-based modeling of cosmogenic nuclides part II–Key aspects of in-situ cosmogenic nuclide production. Quaternary Geochronology, 26, pp.44-55.

*We now state explicitly the two assumptions that underpin our shielding model and cite*

*the Argento et al. paper noting that our method is an incomplete description of the physics in question.*

The point being that the very comprehensive analysis of discretization errors in the shielding calculation here clouds the fact that there exist larger systematic errors due to simplified physics. This issue has basically no practical relevance to the erosion rate calculation overall (because it is still much less important than violations of the basic method assumptions). However, the authors should note here that they are concerned with the precision of a representation of simplified physics, which may or may not be the same as the precision of the calculation relative to real life.

*We now say this so there can be no doubt about what we have done. "Thus our model, while precise, contains a simplified version of the true physics of topographic shielding."*

In addition, this could also be sped up a lot if you really wanted to... but although of historical interest, that is beside the point here.

*Because this is open-source software, future authors can fork our code and make such improvements. Refactoring the code at this point, however, would take several months of effort not only rewriting this component but recalculating every measurement reported here, for minimal gain in accuracy. We don't think the reviewer is asking us to do this so we haven't.*

A final important issue here is that it was not clear to me whether telescoping of the mean free path length on dipping surfaces (see Dunne, also Fig. 5 in Balco, 2014 in Quaternary Geochronology) is included in this calculation.
*It isn't. We say so.*

Line 200 et al. The issue of non-time-dependent vs. time-dependent scaling is actually more important than described here. The reason for that is that production rates are calibrated using data mostly from the last 20,000 years, and the Earth's magnetic field has been stronger than its long-term average during that time. Thus, at erosion rates

low enough that the residence time of material in the soil profile is much longer than this (e.g., most normal erosion rates), the use of a non-time-dependent scheme likely creates a systematic error due to an underestimate of the long-term production rate. Basically this is yet another violation of the steady-state assumption. Again, this is a non-issue compared to much bigger issues in the application of this method, but the text is somewhat inaccurate here as written.

*At the end of the paragraph containing the former line 200, we have inserted a few sentences explaining that for slow denudation rates the assumption of time-invariant production rates will introduce some uncertainty because of the high magnetic field intensity of the past 20 kyrs.*

Line 215, section 2.6. Unfortunately, I simply don't understand why the calculation described in this section is necessary. I didn't look back at the Vermeesch paper, but if I am remembering correctly this whole procedure was just needed to make the equation relating erosion rate to concentration explicit so it could be solved analytically?? Here you don't need to worry about that, because you are only doing the forward calculation, so why are you doing this? I may not be remembering this correctly, but in other words, it seems to me that all the S's and F's needed for Equation 13 are known a priori, or should be. Typically one would compute scaling for spallogenic production and muons separately (for example, this is in the Stone (2000) scheme as published), and because they are different, that should take care of the fact that muons are less important at higher elevation. Each pathway has already been assigned its own attenuation length, so you can compute mass shielding for each pathway. Then it seems like all you need to do is decide how to compute topographic shielding for muons (I don't know the answer...in the 2008 online calculators it is just disregarded), and you are done. What am I missing here? In any case, this needs to be better explained.

*We added two sentences explaining why this is done, but in short, in the equations you have an $S$ term for each production mechanism, but our calculation of the production mechanisms is lumped. So this calculation is to convert lumped scaling terms into four separate scaling terms. There might be a better way to do this, but this way we*

*reproduce cosmocalc exactly, and judging by the minimal differences between CAIRN and CRONUScalc this procedure does not seem to be biasing the results.*

Line 270-ish. I should point out (in response to the editor's comments) that the issue of asymmetry of uncertainty distributions is really a total non-issue from the geological perspective. Anything with an e in it will have an asymmetric uncertainty distribution, of course, but it's hard to think of any cases where it's actually important from the geological perspective.

*This comment will be useful to readers of the discussion, but does not seem to require a change to our text.*

Line 280-ish. Numerical partial differentiation by repeatedly doing the full calculation is almost certainly overkill (especially because you've already linearized it). I would do this simply by assuming that the basin has one pixel with the effective P derived from the whole basin. I agree that it is interesting to do it once, though.

*We agree that it is overkill but we did not know it would be overkill when we were writing the code (we expected the nonlinearities to be larger) and so programmed in the uncertainties the brute-force way. Changing that now would require significant changes to the code, and the uncertainty calculations are not the rate limiting step.*

Line 300. In physical science, 'conservative' is typically used to mean that something is being conserved, e.g., mass or energy. This use in the context of uncertainty analysis is common but incorrect. Instead one should state that the uncertainty estimate is supposed to be an upper bound.

*We have reworded this sentence to reflect that the uncertainty is an upper bound.*

Line 315. This is an excellent point, that the divergence among various theoretical expectations of how snow shielding works is much less important than the practical difficulty of actually measuring the mean snow depth distribution throughout the year. Frankly, in my view it is not even necessary to mention the various models here, because that issue is pretty much totally unrelated to this paper. As an aside, I found the

Delunel paper to not be persuasive because, as far as I can tell from reading the paper, we don't know the effect of snow cover on the energy dependence of the neutron monitors (snow is basically like changing the amount of polyethylene on the outside, which affects the spectral response). This would imply that it is likewise unknown whether or not the variation in monitor count rate with snow cover is applicable to cosmogenic nuclide production at all. But that is totally off topic.
*The other reviewer seems keen on keeping this bit, so we have.*

Line 320-ish. Again, I suggest moving all discussion related to landsliding and non-steady/ nonuniform erosion to a separate section at the end. First, implement the basic model; then, at the end, introduce the abilities to deal with complications that are half-baked at present, but potentially useful in future.
*We have followed this advice and separated the discussion of nonsteady/nonuniform cases from the rest of the model description.*

Line 345-ish. It is probably overkill to generate separate effective elevations and average shielding factors for input to the 2008 online calculator. I know that technically it's required because the elevation affects the muon proportion of total production and the shielding factor doesn't, but this issue is well down in the noise.
*In other cases where we have made a computation that is more robust than really necessary: we don't feel the gain in computational time is worth the effort of refactoring the code at this point. We might do that in a future project to bring the tool online, but feel this level of tweaking is beyond the scope of the current paper.*

Line 370. Again, no need to get into the details of snow shielding here. The point remains that it is unclear whether one can estimate the snow depth accurately in any case.
*We have briefly mentioned the results of Zweck et al here in light of reviewer 1's comments but add no further details.*

In general, in this part of the discussion (i.e., all of section 5) I think it would be easier to

understand if you break down the discussion into two parts: things that are linear with respect to the production rate (e.g., topographic shielding), so can be pixel-averaged by themselves; and things that are nonlinear with respect to the production rate (e.g., elevation, latitude, snow shielding), that have to be converted into production rates, averaged, and then unconverted into an effective summary value. At present these two things are mixed up and it's hard to understand what is happening. Overall, this section could be made more clear.

*We have edited this section and added subheadings to make it more clear. The basic structure remains unchanged because we follow the sequence of calculations that our software actually computes. Firstly, we must calculate self and snow shielding separately, because in CAIRN these are subsumed within the depth-averaging. So we start with a section based on those calculations. We then need to discuss how the different calculators ingest lumped parameters, since they do it differently. Thus we have followed the format of dealing with lumping for each calculator in sequence. To address the comments of the reviewer, we do specifically allude to the nonlinearity and the reason for calculating the effective pressure with the text: "The CRONUS calculators then calculate production using either an elevation or pressure. Production rates are nonlinear with either elevation or pressure, so we must compute an effective pressure that reproduces the mean production rate in the catchment. This is because the arithmetic average of either elevations or pressures within the catchment, when converted to production rate, will not result in the average production rate due to this nonlinearity. CAIRN calculates an effective pressure that reproduces the effective production rate over the catchment. The average production rate is calculated with: ".*

Section 6. This sort of comparison is a terrible mess because of the need to sort out the differences between inherent properties of the algorithm (e.g., point approximation vs. full-basin calculation) and the input parameters (mainly the production rate and the muon interaction parameters). Overall, however, it is accomplished fairly well in this paper; I like the approach of selecting a few representative data sets rather than trying to show a global comparison over all of scaling and erosion rate space, and the

explanation is quite clear as regards which errors apply where in which comparisons. I only have a couple of comments about this section. Line 470. It is quite interesting that there is a systematic difference in shielding factors vs. those originally reported. Do you think this really is because of the averaging a- nonlinear-thing effect? But in any case, as discussed above, the precision of this measurement is overstated in any case due to simplification of the physics.

*We have changed the wording here to simply state that our method produces greater shielding than the other studies. Since the details of those shielding calculations are not reported, we can't exactly diagnose why they are different, so merely mention that our greater shielding values are consistent with the results of our sensitivity analysis on the spacing of azimuth and inclination for the shielding calculation.*

Line 480. I think production rate differences are much more likely responsible here than anything to do with shielding calculations.

*We changed the working so that the production rate differences are mentioned first and topographic shielding is mentioned as a secondary concern.*

Line 486. The snapping issue is by far the biggest problem I can think of in wholesale automation of this process. Especially potentially disastrous for literature data.

*Yes, this is nasty. That is why the repository for CAIRN contains some tools for checking if your sampling point is in the right place.*

Line 500+ and Figures 7-8. As noted by the other reviewer, this effect is nearly all due to differences in the input parameters (production rate and muon interaction cross-sections) and very little due to the spatial-averaging issue or any other aspects of the various algorithms. In large part this is my fault because I have been too much of a slacker to update these parameters in the online code (that is, make v 2.3), which is, frankly, embarrassing. Sorry. However, the need here for the purposes of this paper is to clearly separate these issues. I can think of two ways to do this. One, change the parameters in the CAIRN code to increase spallogenic production by a factor of (4.5/4) and muon production by a factor of (1/0.44). That will very nearly account for the various input parameter differences. Then do the comparison on that basis. Two, leave these figures unchanged but add in the background some lines showing the expected effect of those changes in the parameters (for elevations vaguely resembling the input data). In any case, this would be extremely helpful in distinguishing the various effects of differences in the algorithm itself vs. differences in the calibrated parameters. This would also make the discussion in lines 520+ more clear as well.

*We have followed this advice and generated another figure that shows that the differences between CAIRN and CRONUS2.2 are almost entirely due to the different parameters, and that about a third of this difference is due to the different spallogenic parameters, with the rest being from muons. There still is a small difference between CAIRN and CRONUS2.2 but this is dwarfed by all the other uncertainties.*

1. Reviewer's comment 3. It is not correct to say that 'the model of Heisinger overestimates muon production.' The model accurately estimates muon fluxes; the problem is that the cross-sections for production of Be-10 and Al-26 by muon interactions, when applied to those flux estimates, overestimate Be-10 and Al-26 production. In other words, it's not the model that's wrong, it's the cross-section measurements needed to convert the model prediction to a production rate. I think the present paper is mostly correct on this point.

*See our response to reviewer 1's comment 3.*

2. Reviewer's comment 7. The reviewer is correct here, and this is important. The options here are (i) to get all numbers properly standardized in this discussion, or (ii) preferably, to not get into the details here and simply note that best estimates of production rates are about 10% lower than they were 10 years ago because of improved calibration data.

*We have changed the text following suggestion (ii) above.*

Use of acronym 'CRN.' Do you really want to exclude stable cosmogenic nuclides? Because this code would work for them too. Perhaps, having written the paper, the authors could just globally search and replace 'CRN' with 'cosmogenic nuclide?'

[Figure]

*Done.*

'CAIRN' acronym.

*In our group we amuse ourselves by coming up with acronyms that are Scots words. This is why we eventually rejected the previous acronym frontrunner of Dr NUT (readers can try to guess what that one stands for). Readers can look forward to future models and methods called NUMPTY and GLAIKIT. Readers who disagree with this approach are welcome to come to Edinburgh and discuss the issue over a dram of whisky.*